# Comparing multiple model-derived aerosol optical properties to spatially collocated ground-based and satellite measurements

Ilissa B. Ocko[1] and Paul A. Ginoux[2]

[1]Environmental Defense Fund, New York, 10010, USA
[2]NOAA Geophysical Fluid Dynamics Laboratory, Princeton, 08540, USA

*Correspondence to*: Ilissa B. Ocko (iocko@edf.org)

**Abstract.** Anthropogenic aerosols are a key factor governing Earth's climate, and play a central role in human-caused climate change. However, because of aerosols' complex physical, optical, and dynamical properties, aerosols are one of the most uncertain aspects of climate modeling. Fortunately, aerosol measurement networks over the past few decades have led to the establishment of long-term observations for numerous locations worldwide. Further, the availability of datasets from several different measurement techniques (such as ground-based and satellite instruments) can help scientists increasingly improve modeling efforts. This study explores the value of evaluating several model-simulated aerosol properties with data from spatially collocated instruments. We compare optical depth (total, scattering, and absorption), single scattering albedo, Ångström exponent, and extinction vertical profiles in two prominent global climate models (GFDL CM2.1 and CM3) to seasonal observations from collocated instruments (AERONET and CALIOP) at seven polluted and biomass burning regions worldwide. We find that a multi-parameter evaluation provides key insights on model biases; data from collocated instruments can reveal underlying aerosol-governing physics; column properties wash out important vertical distinctions; and "improved" models does not mean all aspects are improved. We conclude that it is important to make use of all available data (parameters and instruments) when evaluating aerosol properties derived by models.

## 1 Introduction

Industrial, residential, transportation, and agricultural activities have considerably increased the amount of aerosols in the atmosphere since the onset of the Industrial Revolution in the mid-19th Century (e.g., Solomon et al., 2007). Atmospheric aerosols are important for Earth's climate because they are comprised of optically scattering and absorbing particles that can also serve as cloud condensation nuclei (e.g., Boucher et al., 2013). Aerosols' capabilities in either reflecting energy out to space through scattering of sunlight, trapping additional energy in the Earth system through absorption of sunlight and longwave radiation, reducing insolation at the surface, and modifying cloud properties, can significantly alter Earth's radiation budget and influence climate conditions (e.g., Ocko et al., 2014).

Aerosols have relatively short atmospheric lifetimes – on the order of a week – and therefore their atmospheric distributions are relatively localized near emission sources as compared to greenhouse gases. The spatial heterogeneities in aerosol

distribution lead to strong regional differences in radiative forcing, and consequentially in regional climate effects (e.g., Ramanathan and Carmichael, 2008; Shindell and Faluvegi, 2009; Bollasina et al., 2011). Further, sources of anthropogenic aerosols are mainly in the Northern Hemisphere, leading to a meridional asymmetry in distributions and aerosol forcings across the two hemispheres. Aerosol's perturbation of the energy balance specifically in the Northern Hemisphere has been

shown to influence large-scale circulation as well as local climate (Bollasina et al., 2011; Ocko et al., 2014).

Aerosol vertical distributions can also influence climate conditions. Radiative forcings are particularly sensitive to vertical distributions of aerosols due to the relative location of clouds, attenuation of insolation, and relative humidity (e.g., Haywood and Ramaswamy, 1998; Ocko et al., 2012; Samset et al. 2013). The vertical profile of absorbing aerosols, in particular, has a strong bearing on the hydrological cycle (Ming et al., 2010; Ocko et al., 2014).

In order to fully understand how aerosols influence climate, it becomes necessary to employ numerical models to simulate aerosol distributions and properties, evaluate their perturbations to the radiative budget, and investigate changes in thermal, hydrological, and dynamical atmospheric and oceanic properties. To build confidence in model results, however, it is important to evaluate aerosol properties against available observations. For the past few decades, long-term time series measurements of global aerosol properties have accumulated from ground-based and satellite instruments. Spatially

collocated instruments provide opportunities to compare model data with multiple datasets, and the retrieval of multiple aerosol properties from some instruments provides opportunities to evaluate several model-derived aerosol parameters.

The Aerosol Comparisons between Observations and Models (AeroCom) project has pioneered aerosol evaluation in numerous chemistry-transport models and global climate models (e.g., Kinne et al., 2006; Koffi et al. 2012), highlighting model diversity of aerosol properties (e.g., Schulz et al. 2006; Koch et al. 2009; Tsigaridis et al. 2014). Earlier AeroCom

studies relied heavily on two-dimensional AErosol RObotic NETwork (AERONET) and Moderate Resolution Imaging Spectroradiometer (MODIS) measurements (Kinne et al. 2006), while recent studies have incorporated three-dimensional Cloud–Aerosol Lidar with Orthogonal Polarization (CALIOP) measurements (eg., Koffi et al. 2012). However, most studies do not take advantage of all available datasets beyond regional analysis (Kinne et al., 2006; Huneeus et al., 2011), even though a multi-dataset approach can provide a more comprehensive picture (Miller et al., 2011).

Because the horizontal and vertical distributions of anthropogenic scattering and absorbing aerosols dominate a suite of climate responses to the forcings (Ginoux et al., 2006; Donner et al., 2011; Naik et al., 2013; Ocko et al., 2014), it is critical to improve model performance of aerosol optical properties. Here we show that comparing multiple model-simulated aerosol properties – from two prominent, related climate models with vastly different aerosol treatments – to available datasets from spatially collocated ground-based and satellite instruments is important for determining model biases. By characterizing

model strengths and weaknesses, we are able to provide feedback to improve emission scenarios and aerosol properties for future model generations.

We analyze two world-renowned climate models from the same development family – NOAA GFDL CM2.1 and CM3. These models have been used for Coupled Model Intercomparison Project (CMIP) 3 and 5 respectively and are included in the Intergovernmental Panel on Climate Change (IPCC) reports. We build on model evaluations performed by Donner et al. (2011) and Naik et al. (2013) that looked at basic annual-mean aerosol optical properties only. Because the aerosol treatments in the two models are starkly different, as we present in Section 3, comparing multiple optical properties with spatially collocated instruments is especially useful in identifying possible sources of error which are otherwise challenging to determine.

Through evaluation of regional and seasonal model performance using the Multi-angle Imaging SpectroRadiometer (MISR) and MODIS observational datasets, we identify that the largest model discrepancies are isolated to the most polluted areas. We therefore select seven locations worldwide that represent a diversity of conditions, and use high resolution point data (AERONET) and three-dimensional satellite data (CALIOP) to better understand the model biases.

## 2 Observational datasets

We compare present-day model aerosol optical depth (in the visible wavelengths) to satellite observations (MISR and MODIS) to determine regional model performance. We then select seven locations worldwide that (i) have strong model biases based on the MISR/MODIS analysis, (ii) have long-term seasonal time series of measurements (at least seven years of AERONET data), (iii) have relatively large amounts of anthropogenic scattering and absorbing aerosols, (iv) encompass a range of different anthropogenic conditions (such as slightly polluted vs. majorly polluted), and (v) have global coverage; four of the cities represent industrialized regions, while the other three cities represent biomass burning regions. Several model-derived aerosol optical properties at these seven locations are compared to high resolution ground-based data from AERONET and satellite data from CALIOP.

To represent industrialized areas, we chose data from the Atmospheric Radiation Measurement (ARM) Climate Research Facility in Oklahoma, U.S.; Belsk, Poland; Kanpur, India; and Chen–Kung University in Taiwan. The ARM facility, located in the small rural town of Billings, Oklahoma (population around 600), contains the world's largest climate research field site that expands across 9,000 square miles. While it is located in the southern Great Plains, it is not without influence of upwind pollution from heavy industries in Texas as discussed by Andrews et al. (2011). Dust from the southwest U.S., northern Mexico, and even Asia also influence the area (VanCuren and Cahill, 2002; Andrews et al., 2011). Belsk, Poland is a village 28 miles south of Warsaw with a population of less than 10,000. It is influenced by biomass burning in eastern and southern Europe (Jaroslawski and Pietruczuk, 2010) and Saharan dust (Pietruczuk and Chaikovsky, 2012). Poland's shift to a market economy in 1990 and the economic crash of 2008 have reduced pollution over the years (The World Bank, 2011). Kanpur, on the other hand, is one of the most polluted cities in the world with a population of over 2.5 million. The city in India is influenced by heavy industry, nearby deserts, and biomass burning of seasonal agricultural crops (Reddy and

Venkataraman, 2002; Venkataraman et al., 2006; Dey and Di Girolamo, 2010). The local meteorology plays a large role in aerosol load due to suppressed precipitation during the post-monsoon season (Dey and Di Girolamo, 2010). Chen–Kung University is a large, prestigious research university located in Taiwan's oldest city, Tainan (population of nearly 2 million). The area is polluted year-round due to heavy industries nearby (Chen et al., 2009), with seasonal influences from biomass

burning and intense dust storms (Chen et al., 2009; Wong et al., 2013). Of these four sites, the longest AERONET time-series we use is 16 years in Oklahoma (1994–2010), and the shortest is eight years in Belsk and Taiwan (2002–2010).

To represent biomass burning areas, we chose Alta Floresta, Brazil; Mongu, Zambia; and Mukdahan, Thailand. Alta Floresta is located near the Amazon rainforest and is a popular ecotourism destination (population of 50,000). A part of Brazil's "Arc of Fire" – a collection of communities that burn old-growth forests for agriculture or timber – it was in severe violation of

deforestation laws until 2012 (Jackson, 2014). Peak emissions occur in September, although efficient transport of aerosols during the dry season (before August) exports aerosols out towards the Atlantic Ocean (Freitas et al., 2009). Mongu (population around 200,000) is located in tropical, southern Africa along the Zambezi River. Consistent yearly burning of the woody-grassland environment follows a seasonal trend (June through November) characterized by shifts in fuel burnt as the dry season progresses (Eck et al., 2013). Mukdahan in Thailand is also located along a river, Mekong, with a population

around 200,000. Unlike Alta Floresta and Mongu, Mukdahan has two seasonal peaks in aerosol load, one in early spring and one during fall. Nearby crop and vegetation burning, along with wildfires, govern emissions. Alta Floresta contains the longest time-series with 19 years (1993–2012), and Mukdahan contains the shortest with seven years (2003–2010).

### 2.1 MISR

The Multi-angle Imaging SpectroRadiometer (MISR) is an instrument aboard the Sun-synchronous orbiting Terra spacecraft,

and is comprised of nine cameras set at particular angles to capture global multiangle imagery (Diner et al., 1999). Operational since 2000, MISR measures Earth's brightness in four spectral bands (blue, green, red, and near-infrared) and in different directions using a multiangle pushbroom imager. Global coverage time is nine days, with a spatial resolution of 275 m to 1.1 km depending on the channel. MISR has been shown to successfully retrieve AOD over land and water (Kahn et al., 2009), and one of its benefits is its ability to retrieve data over bright deserts. We use Level 3 monthly mean AOD data on

0.5x0.5 degree grid averaged from 2000–2004 to compare with regional model AOD.

### 2.2 MODIS

The Moderate Imaging resolution Spectroradiometer (MODIS) instrument is also aboard the Terra spacecraft (operational since 2000), and complements MISR due to its greater spatial coverage (2330 km wide) and shorter revisit time of two days (Remer et al., 2005). MODIS is a whisk broom imaging scanner with measurements in 36 bands between 0.4 and 14.5 µm.

Studies have found that for coincident retrievals, MISR and MODIS AOD values have a correlation coefficient of 0.9 over

ocean surfaces and 0.7 over land (Kahn et al., 2009). MODIS retrieval uncertainties are found for low AOD where algorithm artifacts are evident. We use the Level 3 monthly mean Collection 6 AOD from MODIS on Terra platform 1x1 degree grid averaged over 2000–2004 (Levy et al., 2013).

## 2.3 AERONET

The AErosol RObotic NETwork (AERONET) is a ground-based, remote sensing, sun photometer measurement network with more than 300 stations worldwide (Holben et al., 1998, 2001). Originally established by the National Aeronautics and Space Administration (NASA) in the 1990s, it has been greatly expanded by other institutions and offers long-term, continuous, and readily accessible data. AERONET provides direct measurements of aerosol optical depth (AOD) and the Ångström exponent (α), and uses inverse algorithms to derive further optical properties such as scattering AOD, absorption

AOD, and single-scattering albedo. There are eight wavelength filters. To isolate AOD from other atmospheric gases and particles, the radiation attenuation due to Raleigh scattering and absorption by ozone and gaseous pollutants is estimated and removed. Three measurements are taken 30 seconds apart, six to nine times a day. We use the Level 2 data, which are quality assured and cloud screened (Smirnov et al., 2000).

We compare AERONET measurements of AOD, scattering AOD, absorption AOD, SSA, and α to the parameters derived by

the CM2.1 and CM3 models for all seven cities, as these locations represent different environments with strong anthropogenic influence. Some data is missing (such as at Alta Floresta) due to lack of high enough AOD for retrieval. The AERONET AOD in the visible spectrum is measured at 440 nm (blue), whereas the model AOD in the visible spectrum is only archived at 550 nm (green) wavelength. Because of the spectral variation of some aerosols across the visible and near-UV spectrum (such as dust) we use the AERONET-measured Ångström exponent between 440 and 670 nm to convert the

AOD at 440 nm data to 550 nm. We use the inverse products derived from the Dubovik algorithm (Dubovik and King, 2000; Dubovik et al., 2002; Dubovik et al., 2006). While AERONET retrievals of AOD are greatly accurate, additional properties derived from inverse algorithms are subject to random noise, systematic errors, instrumental offsets, and uncertainties in the radiation model (Dubovik et al., 2000).

## 2.4 CALIOP

The Cloud–Aerosol Lidar with Orthogonal Polarization (CALIOP) is an instrument aboard the Cloud–Aerosol Lidar and Infrared Pathfinder Satellite Observation (CALIPSO) satellite, launched in 2006 (Winker et al., 2007). CALIOP employs LIDAR to measure vertical profile AOD and extinction at two wavelengths (532 nm and 1064 nm). Global monthly gridded Level 3 data are available from 2007 to 2011 with a vertical resolution of 30-60 m and a horizontal resolution of 333 m. We use the latest version 3 data that have been described and validated by Winker et al. (2013). AeroCom's evaluation with

CALIOP data used Level 2 data from 2007 to 2009 (Koffi et al., 2012). Level 3 provides a more robust comparison than that

by Level 2 due to numerous algorithm improvements, significant bugs fixed, and calibration improvements. CALIOP Level 3 2x5 degree grid has a monthly temporal resolution. Errors in CALIOP data are due to a combination of many factors, such as instrument calibration and offsets, cloud contamination, low signal-to-noise ratio, and uncertainties in multiple scattering, LIDAR ratio, molecular number density, and accumulated aerosol attenuation (Winker et al., 2013). Comparison of CALIOP

AOD with AERONET indicates that CALIOP values are lower, especially at low AOD, due to cloud contamination, scene inhomogeneity, instrument view angle differences, CALIOP retrieval errors, and detection limits (Omar et al., 2013). We compare monthly extinction vertical profile measurements at 532 nm to the model estimates (at 550 nm) for the industrialized and biomass burning sites. While the data we use from CALIOP is spatially collocated with the AERONET stations and model data, it is not temporally collocated. A recent study has shown that temporal collocation can be

significant and sampling errors are introduced when it is not considered (Schutgens et al., 2016).

## 3 Model description and simulations

The National Oceanic and Atmospheric Administration (NOAA) Geophysical Fluid Dynamics Laboratory (GFDL) CM2.1 model (Delworth et al., 2006) and CM3 model (Donner et al., 2011; Griffies et al., 2011) are employed for this analysis. CM2.1 is a state-of-the-art coupled climate model with reasonable aerosol distributions (Ginoux et al., 2006) precomputed

offline through the chemical transport model MOZART-2 (Horowitz, 2006). CM2.1 contains a notably successful simulation of Earth's climate conditions over the past century (Knutson et al., 2006), and is adjudged to be a top tier model based on the climate metric examination by Reichler and Kim (2008). CM3 is the next generation climate model, in which aerosols fields are now calculated online with representations of gas-aerosol chemistry and aerosol-cloud interactions (Donner et al., 2011; Naik et al., 2013).

Aerosol parameters captured by the models include aerosol optical depth (AOD), scattering and absorption AOD, scattering extinction, and absorption extinction. We use these parameters to calculate single-scattering albedo (SSA) and the Ångström exponent ($\alpha$). The Ångström exponent is a proxy of particle size that is derived from simultaneous wavelength measurements of AOD, and relies on the differential measurements to provide an indication of particle size. A smaller $\alpha$ corresponds to larger particles, such as dust. A larger $\alpha$ corresponds to smaller particles, such as black carbon. From analyzing data from 12

cities worldwide, Dubovik et al. (2002) show that $\alpha$ is typically greater than 1 for urban-industrial and biomass burning particles, and typically less than 1 for dust particles. While the model computed extinction and optical depth in the visible and near-infrared wavelength bands, we focus our analysis on the visible wavelengths, which are taken to be 550 nm.

The main aerosol-related differences between CM2.1 and CM3 are (i) aerosols are computed offline (see Section 3.1) in CM2.1 and online in CM3, (ii) emissions inventories are different, (iii) black carbon and sulfate are in external mixtures in

CM2.1 and internal mixtures in CM3, and (iv) the injection height of biomass burning aerosols is included in CM3. CM3 also allows for aerosol-cloud interactions, but we do not consider those here. These changes introduce numerous variables

that make determining discrepancies between the two models' aerosol properties challenging to quantify. Here we compare the aerosol optical properties from both models to form a thorough understanding of what the discrepancies are, building on the initial comparisons provided by Donner et al. (2011) and Naik et al. (2013). We analyze at both regional scales and at key locations (via closest model grid box) where the major discrepancies between observations and models are found. Lack of interpolation of model data in polluted regions may introduce a bias in locations with strong aerosol gradients; however, interpolation is rarely employed for comparisons with observations because the model uncertainties are often larger than the concentration gradient in the grid box. While we analyze the strengths and weaknesses of the models, more research is needed to parse out how individual modifications contribute to the changes in aerosol properties.

## 3.1 GFDL CM2.1

CM2.1 is a coupled atmosphere-ocean-sea ice-land global climate model (Delworth et al., 2006). The atmospheric component, developed by the GFDL Global Atmospheric Model Development Team (GAMDT) in 2004, has a horizontal resolution of 2° (latitude) by 2.5° (longitude), with 24 vertical levels; the top level is around 3 Pa. Aerosol fields are precomputed offline by the global three-dimensional chemical transport model Model for Ozone and Related Chemical Tracers (MOZART-2) (Ginoux et al., 2001; Horowitz et al., 2003; Tie et al., 2005; Horowitz, 2006), with distributions governed by emissions, chemical transformations (i.e. production of secondary aerosols and hygroscopicity), atmospheric transport (advection, diffusion, convection), and wet and dry deposition (Tie et al., 2005, Horowitz, 2006). The CM2.1 model then calculates the aerosol optical and radiative properties online.

Aerosols accounted for in MOZART-2 are sulfate, black carbon, primary organic carbon, secondary organic carbon, and mineral dust (five size bins based on Ginoux et al. (2001)). Aerosol and aerosol precursor emissions are taken from inventories compiled for IPCC Fourth Assessment Report (AR4), and present-day emissions are described in detail in Horowitz (2006). Anthropogenic sources include emissions from fossil fuel combustion, and biofuel and biomass burning. The emissions database used here assumes no seasonality for fossil fuel combustion emissions. Biomass burning, on the other hand, is comprised of a seasonal cycle that is regionally dependent, but is climatological and does not vary year to year. Southern Hemisphere biomass burning sources peak in September-October-November, and Northern Hemisphere biomass burning sources peak in March-April-May. Natural sources, such as wind-driven sea spray and dust, biogenic and soil emissions, background volcanic degassing, and oceanic emissions, remain constant over time. Dust and sea salt emissions are assumed to be entirely natural (Ginoux et al., 2001).

Black and organic carbon are emitted as 80% and 50% hydrophobic, respectively, the rest hydrophilic (Tie et al., 2005), and the hydrophobic compounds are converted into hydrophilic forms with a lifetime of 1.63 days (Reddy and Boucher, 2004). The precursor gas sulfur dioxide is oxidized to sulfate by hydroxyl radical in the gas phase and by hydrogen peroxide and ozone in the aqueous phase, with the reaction rates provided in Tie et al. (2001). Secondary organic carbon is formed via

oxidation of certain volatile organic compounds. Removal parameterizations for dry deposition, gravitational settling, and in-cloud and below-cloud wet scavenging are specified for each aerosol type, described in detail in Tie et al. (2005).

Aerosols are transported by advection, diffusion, and convection according to prescribed meteorological input fields. For all aerosols except dust, meteorological fields by the National Center for Atmospheric Research Community Climate Model (MACCM3) (Kiehl et al., 1998) were employed; dust was simulated separately using meteorological fields from the National Centers for Environmental Prediction (NCEP)–National Center for Atmospheric Research (NCAR) Reanalysis (Kalnay et al., 1996). Sea-salt monthly concentrations are obtained from a previous study by Haywood et al. (1999). They have assumed a surface concentration proportional to the wind speed using the parameterization by Lovett (1978). Sea salt vertical concentration is assumed constant from the surface to 850 hPa, and zero above, and this distribution is kept constant over the years during the simulations (Ginoux et al., 2006).

Horizontal resolution of MOZART-2 is 2.8° by 2.8°, with 34 vertical layers extending up to 40 km (4 hPa). The model time step for chemistry and transport is 20 minutes. Three-dimensional monthly-mean decadal aerosol distributions are archived from MOZART-2 and remapped to the 2° by 2.5° resolution of CM2.1 with 24 vertical levels, and temporally interpolated. Aerosol surface concentrations derived by coupling MOZART-2 and CM2.1 have been thoroughly evaluated by Ginoux et al. (2006).

Aerosol optical depth, single scattering albedo, and asymmetry parameter are calculated from optical properties derived from Mie theory (Haywood and Ramaswamy, 1998) and the concentrations interpolated from MOZART-2 (except for sea salt, which was prescribed following Haywood et al. (1999)). The aerosols are assumed to follow a lognormal size distribution (Haywood and Ramaswamy, 1998). Hygroscopic growth is considered for sulfate (as pure ammonium sulfate modeled after Tang and Munkelwitz (1994), using simulated relative humidity), and for sea salt (as pure sodium chloride (Tang et al., 1997), using a fixed relative humidity of 80%). In the radiative transfer code, black and organic carbon are assumed to remain dry, despite their hydrophilic properties taken into consideration for removal mechanisms in MOZART-2. While anthropogenic fossil fuel emissions of aerosols do not exhibit any seasonality, seasonal humidity generated within the model introduces a seasonal pattern to aerosol optical depths due to hygroscopic growth. Seasonality of aerosol distributions is also influenced by local meteorology.

For our analysis, we use aerosol parameters computed from a 5-member historical simulation ensemble where all forcings vary in time from 1860 to 2000. Five-year monthly mean averages from 1996-2000 are used to represent present-day. We build upon the analysis in Ginoux et al. (2006) that analyzed CM2.1 AOD at 102 AERONET sites and global coverage from satellite data (Moderate Resolution Imaging Spectroradiometer (MODIS)). Ginoux et al. (2006) found that CM2.1 aerosol distributions were often overestimated in polluted regions, and underestimated in biomass burning regions. In this study, we

look at scattering and absorption AOD, scattering vertical extinction, absorption vertical extinction, single-scattering albedo, and the Ångström exponent in addition to overall AOD.

## 3.2 GFDL CM3

CM3 is GFDL's next generation coupled global climate model after CM2.1. Modifications to physics and dynamics are
discussed in Donner et al. (2011). The horizontal domain was changed from a spherical grid to a standard $6 \times 48 \times 48$ cubed-sphere grid, which is effective in avoiding convergence of grid cells at the poles. For reference, the grid boxes for the cubed-sphere framework at the equator are only slightly smaller than that of the Cartesian grid in CM2.1. The amount of vertical levels was doubled from 24 to 48 to better capture stratospheric chemical and dynamical processes, and the uppermost level increased from 3 Pa in CM2.1 to 1 Pa.

The aerosol treatment in CM3 is very different from CM2.1. First, emissions inventories are different. Second, most aerosol distributions (including sulfate, black carbon, organic carbon, dust, and sea salt) are computed online and interactive, such that the distributions are consistent with the model-generated meteorology. Third, aerosol-cloud indirect effects are enabled through the cloud-albedo and cloud-lifetime effects, and the wet deposition scheme is coupled to cloud microphysics (Donner et al. 2011); we note however that we do not include the indirect effects of aerosols in our analysis And fourth,
sulfate and black carbon are assumed to be homogenously internally mixed for radiative calculations.

A modified version of MOZART-2 is inserted into CM3, simulating 97 chemical species, 16 of which are aerosols. Nitrate is simulated in CM3 but is not radiatively active due to its small forcings as found by previous studies (Naik et al,, 2013). However, recent studies have shown that inclusion of nitrate in the radiation scheme improves model AOD (Paulot et al., 2016). Instead of using in-house emissions as in CM2.1 (Horowitz, 2006), the emissions in CM3 were provided by
Lamarque et al. (2010), an inventory that was compiled for the Climate Model Intercomparison Project Phase 5 (CMIP5) in support of the Intergovernmental Panel on Climate Change (IPCC) Fifth Assessment Report (AR5). The emissions of aerosols in Lamarque et al. (2010) are generally lower than that used in CM2.1 (see Table 1); present-day sulfur dioxide, black carbon, and dust emissions are considerably lower in CM3, while organic carbon is higher. As in emissions used for CM2.1, fossil fuel emissions of aerosols contain no seasonal variations. A key improvement to emissions in biomass burning
regions are that the aerosols are vertically distributed, unlike in CM2.1, to more accurately capture the injection height of these aerosols. No information regarding elevation of biomass burning emissions was provided in the inventory, and thus the recommendations of Dentener et al. (2006) are followed to distribute emissions between the surface and 6 km.

As in CM2.1, black and organic carbon are emitted as 80% and 50% hydrophobic, respectively, the rest hydrophilic (Tie et al., 2005). However, the hydrophobic compounds are converted into hydrophilic forms with a lifetime of 1.44 and 2.88 days,
respectively, a change from 1.63 days for both in CM2.1; the increase in lifetime of organic aerosols from hydrophobic to hydrophilic is based on experimental evidence in Huang et al. (2013), which uses a process-based aging scheme including

the effects of chemical oxidation and physical condensation/coagulation. The treatments of chemistry and deposition are similar in CM2.1 and CM3 (Donner et al., 2011). Secondary organic aerosols (SOA) are formed by the oxidation of non-methane volatile organic compounds (NMVOCs) from both natural and anthropogenic sources, as described by Dentener et al. (2006) and Tie et al. (2005), respectively; SOA from natural sources is produced by rapid oxidation of biogenic terpenes,

and SOA from anthropogenic sources is produced by OH-induced oxidation of butane. Transport of aerosols is similar overall, but there are differences in large-scale and subgrid transports that are responsible for some changes in aerosol fields. Recall that aerosols are now interactive within the meteorology of the model.

Calculations of aerosol optical properties (aerosol optical depth, single scattering albedo, and asymmetry parameter), size distribution assumptions, and refractive indices are unchanged from CM2.1 to CM3. Lognormal size distribution is assumed

for sulfate and carbonaceous aerosols, also unchanged from CM2.1. However, hygroscopic growth was limited to 98% relative humidity in CM3 rather than 99.9% in CM2.1, as 99.9% was shown to produce excessive AOD in CM2.1 (Ginoux et al., 2006). Further, a key inclusion in CM3 is a globally pervasive internal mixing assumption that considers a homogenous mixture between sulfate, black carbon, and water by a volume-weighted average of their refractive indices. As in CM2.1, aerosol AOD exhibits seasonality in part due to the seasonal variation in local relative humidity, despite fossil fuel emissions

not varying seasonally.

We use aerosol parameters computed from a 5-member historical simulation ensemble where all forcings vary in time from 1860 to 2005. Five-year monthly mean averages from 2000-2004 are used to represent present-day. We build upon the analyses in Donner et al. (2011) and Naik et al. (2013) that compared model-derived AOD to observations. Donner et al. (2011) found that while the emissions of black carbon are considerably decreased from CM2.1 to CM3 (Table 1), changes in

AOD are partly compensated by increased absorption from internal mixing with sulfate. Further, reduced aerosol direct effects in CM3 led to increases in clear-sky downward shortwave radiation that were more consistent with observations, providing strong evidence that aerosol direct effects are better represented in CM3 than in CM2.1 (Donner et al., 2011). Naik et al. (2013) find that the mean bias of the CM3-simulated global aerosol optical depth is within 5% of satellite measurements over 1982 to 2004. Of the years in which volcanic aerosols in the atmosphere represents a minor contribution

(1996 to 2006), the mean bias is within 2%. Overall, the improved AOD in CM3 is attributed mostly to changes in emissions and the new internal mixing treatment (Donner et al., 2011). We extend these previous evaluations of model performance by expanding the amount of aerosol properties compared to observations, and also analyzing the vertical extinction distributions. We are therefore able to offer further insight into the discrepancies such that future model generations can improve their treatment of aerosols.

**4 Results and discussion**

## 4.1 Model aerosol properties

Fig. 1 shows a comparison between total aerosol optical properties simulated by the models – aerosol optical depth (AOD), aerosol absorption optical depth (AAOD), and single-scattering albedo (SSA), and Table 1 presents the global-mean values for the total optical depths. CM2.1 values are averages from 1996–2000, and CM3 values are averages from 2000–2004 as these are the closest model years to "present day" conditions. We do not expect the different years to be an issue because of the five-year monthly averages and absence of any major aerosol events like volcanic eruptions.

The applied emissions and resulting global aerosol burden in CM3 is significantly lower than that by CM2.1 (Naik et al., 2013). However, while sulfate and black carbon emissions and burdens are considerably lower in CM3, the overall aerosol optical depths (total, absorption, and scattering) are very consistent between the two models (Table 1). While reduced emissions and a lowered cap of relative humidity for sulfate hygroscopic growth yield a reduction in AOD in CM3, internal mixing of black carbon and sulfate introduced in CM3 produces higher AAOD than CM2.1, as explained by Persad et al. (2014). Further, organic carbon and sea salt global-mean optical depths have slightly increased from CM2.1 to CM3 (Fig. 2).

While the global-mean AOD and AAOD are relatively consistent between the models, the spatial distributions show considerable differences. AOD and AAOD over Northeast U.S., Europe, and Australia source regions are much lower in CM3, whereas Brazil, Indonesia, and India show much higher AOD and AAOD in CM3. It is also clear that more scattering and absorbing aerosols are penetrating into the Arctic in CM3 than in CM2.1. The SSA plots show a higher SSA globally in CM3, (indicative of more scattering aerosols relative to absorbing), and particularly evident over Brazil and the Sahara desert. Part of the discrepancy is related to the difference in climate meteorology simulated by the two models (Donner et al., 2011).

Figure 2 breaks down the total AOD into individual components, and Table 1 provides global-mean emissions, burden, and optical depths for all radiatively active aerosol species. While sulfate and black carbon are internally mixed in CM3, the total extinction is partitioned between the two species based on mass.

The individual aerosol AOD differences between CM2.1 and CM3 (Fig. 2) explain several regional differences seen in Fig. 1. Black and organic carbon biomass burning regions in South America, Africa, and Asia dominate their respective AODs in CM3, with North America and Europe playing minor roles. Sulfate is constrained in closer proximity to sources in CM3, yielding less diffusion of AOD in the Northern Hemisphere. Dust from the Sahara plays a slightly lesser role in CM3 than CM2.1 due to lower emission. Sea salt's regional pattern and magnitude is improved from the earlier version for which it was prescribed with constant value below 850 mb.

Overall, attributions of aerosol species to total global-mean AOD are 5, 59, 6, 18, and 12% in CM2.1 for black carbon, sulfate, organic carbon, dust and sea salt, and 2, 43, 19, 11, 25% in CM3, respectively.

**4.2 Comparisons of model data with observations**

**4.2.1 Regional comparisons**

First we compare regional model monthly mean AOD with satellite data from MISR and MODIS (Figs. 3, 4, and 5). Both CM2.1 and CM3 successfully reproduce the magnitudes of AOD in unpolluted regions (Fig. 3). In polluted regions in the
Northern Hemisphere mid-latitudes, there is a great improvement in AOD magnitude from CM2.1 to CM3 despite lower emissions (Fig. 3). CM2.1 AOD is essentially dominated by sulfate everywhere but Northern Africa, with too little organic and black carbon in tropical regions and sea salt in the Southern Ocean (Fig. 4). The CM2.1 biases have been related to inadequate parametrization of emission for sea-salt and carbonaceous aerosols and excessive hygroscopic growth for sulfate (Ginoux et al., 2006). With improved emissions of sea-salt and biomass burning aerosols, and reduced sensitivity to
hygroscopic growth in CM3 (Donner et al., 2011), Fig. 5 shows a larger diversity of aerosol types to AOD compared to CM2.1 (Fig. 4). CM3 seasonal AOD is mostly controlled by the sulfate summer maximum in the Northern Hemisphere mid-latitudes, dust in the subtropical regions, biomass burning in the tropics, and sea salt in the Southern Ocean (Fig. 5).

However, unlike the magnitudes, the models' abilities to capture seasonality of AOD worsen from CM2.1 to CM3, except in the tropics (Fig. 3). Global dust emission has been uniformly reduced in CM3 compared to CM2.1 (Donner et al., 2011),
which is reducing its major contribution to the spring maximum over East Asia as well as over the North Pacific (Fig. 5) (Yu et al., 2012). Previous studies have shown that inclusion of nitrate in CM3 considerably improves the seasonal cycle by reducing the contribution of sulfate to AOD in summer while increasing it in winter (Paulot et al., 2016).

All of these elements participate to deteriorate AOD seasonal variation from CM2.1 to CM3 on a regional scale. In Sect. 4.2.2 we will analyze these biases in more detail by focusing on key regions that represent a diversity of locations and
pollution sources (Fig. 6): Oklahoma, U.S.; Belsk, Poland; Kanpur, India; Chen–Kung University in Taiwan; Alta Floresta, Brazil; Mongu, Zambia; and Mukdahan, Thailand.

While MISR and MODIS are useful for analyzing a broad global coverage of AOD, high resolution data is necessary for point analysis. Therefore, we employ AERONET and CALIOP to evaluate model data at the seven key locations as they are complementary in that AERONET provides data for multiple aerosol parameters (e.g. aerosol optical depth (AOD), aerosol
absorption optical depth (AAOD), single-scattering albedo (SSA), and the Ångström exponent ($\alpha$)) and CALIOP provides vertical extinction profiles. While studies often compare one or two parameters (e.g. Kinne et al., 2006), the availability of multiple parameters is valuable in evaluating aerosol properties in a model. Further, spatially collocated instruments are beneficial in understanding the discrepancies between model and observations.

**4.2.2 Evaluating multiple aerosol parameters in polluted regions**

Comparisons of the five-year averages of model data (CM2.1 and CM3) with averages of all available AERONET data are found in Figs. 7 (polluted cities) and 8 (biomass burning regions). Aerosol parameters compared include AOD, scattering AOD, AAOD, SSA, and $\alpha$. The error bars for the AERONET data represent year-to-year variability in the available data. Correlation coefficients for monthly mean model versus AERONET data is shown inset.

The site in Oklahoma is in a rural environment compared to the other urban sites we have chosen for model evaluation, and therefore represents areas with background pollution. As expected, total AOD is considerably lower than the other sites. The largest nearby cities are Wichita, Kansas; Tulsa, Oklahoma; and Oklahoma City, Oklahoma; and each is at least 100 km away. Pollution sources may be heavy industries in Texas cities such as Houston and Dallas, while dust from the southwest U.S. and northern Mexico, and possibly long-range transport of dust from Asia (VanCuren and Cahill, 2002; Andrews et al.,

2011) could also be causal factors. Air mass back-trajectories show that summertime air originates in polluted regions of Texas, while wintertime air is from cleaner, northern sources (Andrews et al., 2011).

Both CM2.1 and CM3 reproduce the AOD (total, scattering, and absorption), SSA, and $\alpha$ magnitudes to well within a factor of two (Fig. 7). The seasonality is adequately represented, and is very strong in CM3 (CM2.1 $r^2 = 0.70$, CM3 $r^2 = 0.97$). AERONET shows an Oklahoma AOD maximum in May and August, while CM2.1 derives peaks in April and September

attributable to high sulfate AOD (not shown). AERONET, CM2.1, and CM3 all show AOD minimums from October to February. The models and AERONET both show a drop in $\alpha$ during springtime reaching a minimum in April, which is indicative of the presence of dust, and a rebound in summer attributed to the peaks in sulfate.

CM2.1 AOD is considerably overpredicted in Belsk by a factor of three or more from April to September.This is partly attributed to Poland's shift to a market economy in 1990 that has since steadily reduced pollution emissions (The World

Bank, 2011) and partly attributed to the economic crash of 2008, not considered in the model. The discrepancy may thus be attributed to a mismatch of periods between the model and AERONET. CM3, however, represents the AOD magnitudes in Belsk to within a factor of two.

AERONET shows one peak in April, and another in July–August. CM2.1 reasonably reconstructs the seasonality ($r^2 = 0.45$) with a slight peak in AOD in April, slight dip May to June, and then slight peak again in July to September, whereas CM3

only has one peak in June and an $r^2$ of 0.15. Analysis of back-trajectories computed using the NOAA HYSPLIT model and fire maps show that these peaks coincide with seasonal biomass burning in eastern and southern Europe (Jaroslawski and Pietruczuk, 2010). Another study uses LIDAR measurements and model results to suggest that transport of Saharan dust also influences springtime AOD in Belsk (Pietruczuk and Chaikovsky, 2012). This is consistent with CM2.1 (and not CM3), which shows a maximum absorption AOD, minimum SSA, and minimum Ångström exponent during March–April–May, as

well as a peak in the dust AOD in May. CM3 does, however, capture the seasonal variation in SSA and particle size

(indicative of the seasonal mix of aerosols) extremely well (SSA $r^2 = 0.69$, α $r^2 = 0.94$), even though the AOD seasonality is poor (AOD $r^2 = 0.15$).

For Kanpur, peaks in AOD during May and October are partly associated with peaks in open biomass burning of rabi and kharif agricultural crops, respectively (Venkataraman et al., 2006). There is also a significant enhancement in dust loading during the pre-monsoon season (April to June) (e.g., Ginoux et al., 2012). Post-monsoon, aerosols transported to or emitted near Kanpur can accumulate rapidly in the atmosphere from suppressed precipitation (Dey and Di Girolamo, 2010). Monthly emissions from fossil fuel and biofuel combustion are fairly constant (Reddy and Venkataraman, 2002). Dey and Di Girolamo (2010) analyzed nine years (2000–2008) of AOD seasonal climatology derived from the MISR – an instrument aboard the NASA Terra spacecraft – and the results are consistent with those shown by AERONET. Dey and Di Girolamo (2010) used air mass back trajectories – calculated using the NOAA HYSPLIT model – to show that the Great Indian Desert and the Arabian Peninsula are the likely sources of the dust.

CM2.1 AOD is consistently underpredicted in Kanpur by a factor of four on average, but this is expected because Kanpur has incredibly high pollution levels of which models with the resolutions of CM2.1 and CM3 are not expected to resolve. CM3 does, however, show improved magnitudes. The CM2.1 and CM3 maximum AOD coincides with a minimum AOD as measured by AERONET during the monsoon season (July to September) resulting in negative correlation coefficients. Likewise, the CM2.1 and CM3 minimum AOD coincides with a maximum AOD as measured by AERONET during the dry season (October to January). The presence of high AOD in the winter months is verified by several satellite instruments including CALIOP (Cloud–Aerosol Lidar with Orthogonal Polarization), MODIS (Moderate Resolution Imaging Spectroradiometer), MISR (Multiangle Imaging Spectroradiometer), OMI (Ozone Monitoring Instrument), and TOMS (Total Ozone Mapping Spectrometer) (Ganguly et al. (2009b) and references therein).

Emissions inventories in India have large uncertainties (Venkataraman et al., 2006), and because both CM2.1 and CM3 do not prescribe any seasonality in emissions from anthropogenic sources, it is unsurprising that the chemistry–transport or chemistry–climate models cannot reconstruct Kanpur's AOD seasonal climatology. Ganguly et al. (2009b) found that a decoupled version of the model used here (GFDL AM2) largely underestimated carbonaceous aerosols in the Kanpur region by as much as a factor of 10 during winter months. The high summer bias in Kanpur AOD during the summer months is likely due to convective removal of aerosols simulated too low, therefore leading to high biases especially in the tropics where convective large scale precipitation is dominant (Paulot et al., 2015). Further, in addition to dust transport from desert regions, anthropogenic sources of dust are prevalent in India from agricultural activities and land use (Ginoux et al., 2012). This is also not accounted for in CM2.1.

SSA observations are much lower in Kanpur as compared to the other sites (~0.88 compared to ~0.92–0.98), which is representative of its relatively large black carbon and dust concentrations. Kanpur also has the smallest α of the four sites.

Both models show this as well. Kanpur α is consistent with the observation that the pre-monsoon aerosol loading includes a large dust component. While CM2.1 underestimates AOD in Kanpur and does not simulate the seasonal climatology, the absorption AOD and α suggest that dust particles are present in the model from March to May. CM3 overpredicts α in Kanpur by at most a factor of three, perhaps showing too little dust.

Aerosols in Taiwan have industrial, biomass burning, and dust storm sources. The region is highly polluted from nearby heavy industries year-round (Chen et al., 2009). Springtime aerosols in Taiwan and Southeast Asia are partly attributed to intense dust storms in Mongolia and North China, which have been observed to travel in the mid-troposphere all the way to Europe (Grousset et al., 2003), and have also been observed at lower latitudes in Taiwan (Chen et al., 2009; Wong et al., 2013). Peak biomass burning season in Southeast Asia also occurs during the spring (Streets et al., 2003). CM2.1 somewhat

captures the March–April peak, although CM3 shows a peak during summer months where AOD is at a minimum; this may also be due to the low convective removal of aerosols leading to high biases in the tropics (Paulot et al., 2015). CM2.1 also accurately captures the October–November peak and summertime minimum ($r^2 = 0.43$), whereas CM3 does not ($r^2 = -0.45$).

Both the dust and carbonaceous aerosols from biomass burning likely contribute to a drop in the SSA during the springtime months (seen in AERONET and models), although dust is more absorbing in the near-UV than in the visible spectrum (Giles

et al., 2012). The lack of a drop in α derived from AERONET during spring likely represents the balancing out of more large particles (dust) and more fine particles (black and organic carbon).

In autumn, air mass back trajectories computed by Chen et al. (2009) using the NOAA Hybrid Single Particle Lagrangian Integrated Trajectory Model (HYSPLIT) model show variable sources including northwest China, southern China, and the Pacific Ocean. Therefore, it is hard to attribute the cause of the secondary peak in AOD during October–November.

Overall, CM2.1 and CM3 satisfactorily reproduce AOD magnitudes in the key industrial regions, with an improvement from CM2.1 to CM3 – as shown for most regions worldwide in Section 4.2.1 (Fig. 3). The decline in performance of CM3 AOD seasonality is clear for the most polluted regions (Belsk, Kanpur, and Taiwan) as there is a large drop in the correlations from CM2.1 to CM3. While the CM3 AOD seasonality is worse than CM2.1, this is not true for the reproductions of the single-scattering albedo and Ångström exponent. Using CM2.1 dust and black carbon absorption AOD as a proxy for

concentrations, peaks in CM2.1 dust absorption AOD are correlated well with dips in the AERONET α, and peaks in CM2.1 black carbon absorption AOD are correlated well with peaks in the AERONET α (not shown).

The model suggests that, for the industrialized sites, the total and scattering AODs are dominated by sulfate, and absorption AOD is dominated by black carbon with a significant contribution from dust (not shown). Organic carbon, and sea salt especially, play minor roles at most. However, Heald et al. (2005) suggest that global climate models underestimate the

contribution of organic carbon to the total aerosol concentration and AOD. In particular, Ganguly et al. (2009a) suggest that sulfate concentrations are overestimated and organic concentrations are underestimated in CM2.1 over Oklahoma.

### 4.2.3 Evaluating multiple aerosol parameters in biomass burning regions

For the biomass burning regions (Fig. 8), the models' AOD magnitudes are much less consistent with the AERONET observations during burning season. CM2.1 consistently underpredicts total AOD by a factor of 4–6 during peak biomass burning emissions (September for Alta Floresta and Mongu, and March for Mukdahan). CM3 shows similar results for Alta Floresta and Mongu, but with a huge spike in Mukdahan AOD during June through September that rivals March magnitudes, a time when AOD is at a minimum based on AERONET measurements. The deterioration of seasonality from CM2.1 to CM3 is shown clearly for AOD in all biomass burning sites (e.g. Alta Floresta CM2.1 $r^2 = 0.94$, CM3 $r^2 = 0.07$), although α seasonality is consistent across models, and even improves for Alta Floresta (CM2.1 $r^2 = 0.90$, CM3 $r^2 = 0.95$).

Underestimations during peak biomass burning season may be due to underestimated emissions, an injection height that is too low, efficient wet removal in convective regions, and/or the lack of hygroscopic growth of carbonaceous areosols. The severe underestimates in biomass burning aerosols in the models could impact model-derived climate changes important to understanding aerosol's role in climate change, due to the lack of aerosols in the Southern Hemisphere which would play a role in cross-equatorial energy balance (Ocko et al., 2014).

Although AOD are underestimated during peak biomass burning season, there is a slight peak in model AOD (with the exception of Mukdahan) suggesting that the model does capture the seasonal cycle of biomass burning, just not the magnitude of the emissions or concentrations. The models do, however, include a secondary peak in Mukdahan emissions in September–October, and the AOD magnitudes are consistent as well. When it is not biomass burning season in any of these regions, CM2.1 is consistent with AERONET observations, although CM3 shows higher AOD from December through March in Alta Floresta and Mongu that has no parallel in the AERONET data.

Alta Floresta AERONET AOD maxima are the highest of any comparison site analyzed in this study. As shown in Fig. 8, AOD during the main biomass burning season in Alta Floresta (August to September) has large error bars, but this is due to severe deforestation at the beginning of the dataset which later declines significantly when the Brazilian federal government cracked down on deforestation violations starting in 2008 (Jackson, 2014). The average AOD from 1993–2012 in September is almost 1.5. In fact, Hoelzemann et al. (2009) found that Alta Floresta had the highest AOD observed (4.0) of all 12 observation sites the study analyzed in South America using MODIS satellite data. The large AOD is due to intense fire activity that exists in the vicinity of Alta Floresta due to deforestation. Prior to August, during the dry season, climatological patterns in central Brazil may efficiently dilute pollution by exporting it to the ocean (Freitas et al., 2009).

Over Alta Floresta, AERONET derives a SSA for July through December of approximately 0.95. CM2.1 projects a sharp decline in SSA during these months that drops to 0.8, but CM3 brings the magnitudes back up closer to observations. While absorption AOD in the models is only underestimated by a factor of 2 in Alta Floresta during peak emissions (September), scattering AOD is underestimated by more than 4 times. This means that the model accounts for more absorption relative to

scattering than is actually present, yielding low SSA. A larger scattering AOD would yield SSA closer to 1. The Ångström exponent derived by AERONET is consistent with CM2.1, and shows a drop in α from January to May; CM3 also shows this pattern, but overestimates the amount of smaller particles. During peak fire activity, however, α is considerably higher. There is also large year-to-year variability in α in Alta Floresta, which may also be due to reduced emissions in later years.

Maximum seasonal AOD in Mongu as measured by AERONET from 1995 to 2010 is half that of Alta Floresta (0.8).

Tropical Africa is characterized by widespread and frequent forest fires that occur consistently each year. MODIS satellite data show that burning begins in May and peaks July to September (Giglio et al., 2003, 2006). This is slightly shifted from AERONET AOD data, which show maxima between August and October. The apparent offset between MODIS fire activity and AERONET AOD is corroborated by aircraft data analyzed in Magi et al. (2009), which shows a shift in peak AOD by 1– 2 months after peak fire activity. CM2.1 and CM3 show slight increases in AOD that align in timing with AERONET,

although CM3 has an additional peak in February which is not present in the observations.

As mentioned earlier, the models capture one of the two AERONET AOD maxima over Mukdahan. Mukdahan is influenced by nearby crop and vegetation burning and wildfires throughout the dry season; biomass burning activity first peaks in March, with a second, smaller peak in autumn after the rainy season (Boonjawat, 2008). The February to April peak evident in AERONET in fact corresponds to a decrease in model AOD. The September–October peak, on the other hand, is well

captured by CM2.1 in both timing and magnitude. The springtime AOD in the observations is considerably higher than that in autumn, which may be a result of changes in burning conditions (wildfire vs. controlled burn) and vegetation type (Dubovik et al., 2002). Concurrently, the AERONET data show a drop in SSA and a peak in α that is extremely consistent with the models and suggestive of carbonaceous particles from burning. Because of the lack of moisture during the winter dry season, the peak in α may also be attributed to the lack of hygroscopic growth (Logan et al., 2013). The models also

show a slight increase in dust AOD during summer in Mukdahan (not shown), which matches up well with a sharp dip in AERONET α.

Overall, model AOD magnitudes are more consistent with AERONET observations in industrialized areas than for biomass burning areas, although the model reproduces satisfactorily SSA and α at most sites. Model AOD magnitudes in biomass burning regions are less consistent with the AERONET observations, and maximum AOD are underestimated in the model

by as much as factors of 4 to 6. While CM2.1 captures the seasonality for AOD for most locations, CM3 shows peaks when there are minimums. While some of the discrepancy between CM2.1 and CM3 is due to different meteorology (Donner et

al., 2011), differences between model and observations also arise because the climate models are unable to reproduce specific synoptic events.

### 4.2.4 Evaluating model-derived data with spatially collocated instruments

Whereas AERONET provides a two-dimensional view of aerosol properties via total column estimates, CALIOP measurements provide insight into the vertical structure of the aerosol properties, revealing the elevations of aerosols. This is incredibly important and useful for climate model evaluation because aerosol radiative effects are extremely sensitive to elevation (e.g. Ocko et al., 2014). Satellite measurements are further valuable because they have a broad spatial coverage. Here we weave in analysis of CALIOP data to the existing AERONET/model discussion. Figs. 9 and 10 compare the seasonal CALIOP measurements at 532 nm to the models' estimates (at 550 nm) for the industrialized and biomass burning sites, respectively.

As discussed earlier, CM3 captures the AERONET AOD seasonality for Oklahoma extremely well ($r^2 = 0.97$) and better than CM2.1 ($r^2 = 0.70$). However, when comparing the vertical distribution of extinction over the ARM facility using CALIOP measurements, CM2.1 outperforms CM3 because of its ability to capture the vertical structure of the seasonality. On the other hand, CM3 does not exhibit two distinct seasonal peaks at higher elevations as shown in both the CALIOP and CM2.1 data. This highlights the need for and value of comparing model data to multiple observational datasets.

For Belsk and Taiwan, CM2.1 reproduces the seasonality and elevations of extinction, although surface extinctions are overestimated at both sites (by a factor of five in Belsk during summer). CM3 more accurately captures surface extinction magnitudes in Belsk and Taiwan (overestimate by a factor of two in Belsk during summer) even though it completely fails at reproducing the seasonality (Belsk $r^2 = 0.15$; Taiwan $r^2 = -0.45$). As discussed previously, emissions in eastern Europe were considerably reduced as a result of switching economic regimes as well as subsequent economic stress from the crash of the world economy.

Belsk maxima extinction in CALIOP are offset from AERONET data (March instead of April, October–November instead of August–September). However, AERONET does show a peak in absorption AOD in March with high year-to-year variation (large error bar), as well as a drop in both SSA and α. This is consistent with conclusions by Pietruczuk and Chaikovsky (2012) that dust is transported to Europe from the Saharan desert during spring. As shown before on a regional scale, CM3 improved the magnitudes considerably, but the seasonality deteriorated.

Taiwan is the only site where AERONET suggests considerably higher total AOD than the vertically-integrated CALIOP data (not shown). Both CM2.1 and CM3 models have magnitudes more consistent with AERONET than CALIOP, although they both underestimate the springtime maxima by a factor of 1.5. However, while the AERONET comparison suggests that CM2.1 underestimates extinction during the peak season, comparison of the vertical extinction profile shows that CM2.1

extinction is constrained at the surface and considerably larger than that of CALIOP by over a factor of four; on the other hand, CALIOP data shows that the extinction profile extends up to 4 km in elevation (similar magnitudes of extinction in CM2.1 only reach 2 km.) While the springtime AOD peak in AERONET is slightly larger than the peak in autumn, CALIOP shows large differences in the vertical distribution of extinction during spring and autumn. During springtime, aerosol extinction reaches higher elevations than during autumn. This is consistent with studies showing long-range high-elevation transport of dust (Lin et al., 2007), and also consistent with ground-based LIDAR measurements in Taiwan (Chen et al., 2009). Recall that the main springtime sources of aerosols in Taiwan (other than industry) are dust transported from the north, and nearby biomass burning. During autumn, high extinctions are constrained closer to the ground. Interestingly, the total column optical depth when computed from CALIOP data show that the overall AODs are similar for spring and autumn, even though their vertical distributions vary tremendously. This shows the value of instruments like CALIOP in their ability to resolve aerosol vertical profiles. The model, on the other hand, does not accurately distinguish the differences in the vertical profiles over Taiwan, and springtime and autumn extinction distributions are fairly comparable.

For Kanpur, both models capture the fall peak but not the spring peak. In contrast to CM2.1 performance in Belsk and Taiwan (overestimated surface extinction magnitude), CM2.1 considerably underestimates the magnitude and elevation of extinction in Kanpur. CM3, on the other hand, captures the magnitudes and elevation. As discussed in Sect. 4.2.2, the wintertime and pre-monsoon seasons are largely influenced by enhanced dust loading from nearby deserts, and agricultural and land use activities (Ginoux et al., 2012). While the CM2.1 model accounts for natural sources of dust, and captures a slight peak in dust emissions during this period (Fig. 7), it does not account for anthropogenic sources. It is very likely that CM2.1 also underestimates concentrations of carbonaceous aerosols from biomass burning in this region during these seasons. On the other hand, AERONET and CALIOP show the same seasonal trends, except that AERONET AOD remains high during winter months whereas CALIOP extinction drops from November to February.

Fig. 10 compares the seasonal CALIOP measurements at 532 nm to the model estimates (at 550 nm) for the biomass burning sites. Extinction seasonality shown by CALIOP is consistent with AERONET AOD, with maximum extinction during September in Alta Floresta and Mongu, and March/September–October for Mukdahan. It is clear from the CALIOP data that aerosols reach much higher elevations over biomass burning influenced sites as compared to industrialized sites; the strongest extinctions can extend up to 3 km in the atmosphere.

As suggested by Fig. 8 in the AERONET comparisons, CM2.1 and CM3 completely miss the large magnitudes of extinction during biomass burning seasons, with the exception of CM3 Mukdahan in the fall. While the AERONET and model correlation coefficients suggest that only CM2.1 Alta Floresta and Mongu seasonality is captured, analysis of CM2.1 and CM3 vertical extinctions show that the models do capture the seasonal cycles at all three sites, but with extinctions that are underestimated by a factor of 10. Higher extinction magnitudes are also constrained closer to the surface in the models, whereas in reality, and as shown by CALIOP, aerosols from biomass burning sources can be lofted high into the atmosphere.

In CM2.1, carbonaceous aerosols from biomass burning are emitted from the surface (Horowitz, 2006). However, because it has been known for quite some time that injection height from open fires and wet deposition are key to simulating smoke plumes properly (e.g., Westphal and Toon, 1991), biomass burning emissions for CM3 were distributed vertically following the recommendations of Dentener et al. (2006) (distributed emissions between the surface and 6 km). As discussed before,

aerosols in the boundary layer have a relatively short lifetime due to efficient dry deposition at the surface by the turbulent boundary layer, while aerosols injected into the mid- or upper troposphere can be transported over very long distances. Our results show that CM3 likely needs to increase the vertical structure for the biomass burning emissions, because particles are still not distributed high enough when compared to CALIOP data. However, magnitudes for Alta Floresta and Mongu need to be higher in addition to modifying the injection height.

Other factors that may contribute to the underestimate in concentrations over biomass burning regions are underestimated emissions, too much wet removal in convective areas, the ratio of hydrophilic to hydrophobic aerosols, vertical mixing (convection), and hygroscopic growth of carbonaceous aerosols.

Overall, comparing the models vertical profiles of extinction with CALIOP data for all seven sites shows that seasonality is reproduced much better than the magnitudes and elevations of extinction, both of which are controlled by meteorology and

emissions in the case of biomass burning sites. It is interesting that the models can reproduce seasonality in industrialized regions when emissions do not have a prescribed seasonal distribution. For biomass burning sites in particular, which are controlled by emissions inventories, seasonal maxima are temporally consistent, however, the extent and height of aerosols in the atmosphere is severely underestimated (except for Mukdahan in late summer). This problem may be attributed to a combination of factors, such as the lack of modeling of the injection height of particles from open fires (and therefore

efficient removal in the turbulent boundary layer), underestimated emissions, or excess wet removal of aerosols in convective regions. AERONET and CALIOP measurements are fairly consistent with one another, and show similar seasonal patterns. This is a first step towards understanding the model biases, and more research is needed to parse out individual causes.

**5 Conclusions**

The GFDL CM2.1 and CM3 global climate models are world-renowned, used for CMIP 3 and 5, and are included in the IPCC reports. While these models are from the same development family, aerosol treatment is starkly different due to updated emissions inventories and model improvements such as interactivity with meteorology and clouds, internal mixtures, and accounting for biomass burning injection heights. The myriad aerosol changes from CM2.1 to CM3 make evaluation of aerosol performance challenging when considering future improvements.

Donner et al. (2011) evaluated basic aerosol-related properties (annual-mean AOD, coalbedo, clear-sky downward shortwave radiation) of CM2.1 vs. CM3, leading to the general conclusion that the direct effects of aerosols are more realistically simulated in CM3. Naik et al. (2013) further evaluated CM3 aerosol optical performance to find that global annual AOD is within 2% of satellite measurements over 1996 to 2006.

We build upon these previous studies by comparing multiple monthly-mean model-derived aerosol optical properties with observations from four measurement techniques – satellite imagers (MISR and MODIS), ground-based sun photometers (AERONET), and satellite LIDARs (CALIOP). While we find that AOD magnitudes do improve from CM2.1 to CM3 on a regional scale, seasonal variations are better simulated by CM2.1. Because the major biases are found in polluted regions, we select seven sites with industrial and biomass burning sources of aerosols to analyze the biases in more detail by comparing

multiple aerosol properties and employing spatially collocated instruments. The sites include an urban-influenced rural area in Oklahoma, USA, with industrial and dust sources; Belsk, Poland, with industrial and dust sources; Kanpur, India, with industrial, dust, and biomass burning sources; Taiwan with industrial, dust, and biomass burning sources; Alta Floresta, Brazil with biomass burning sources; Mongu, Zambia, with biomass burning sources; and Mukdahan, Thailand, with biomass burning sources. We have also compared our results to several previous studies – modeling, observational, or both –

for each comparison site.

Comparing multiple aerosol optical properties derived by models to measurements from collocated instruments both identifies opportunities for the improvement of modeling aerosol distributions, as well as reveals important aspects governing aerosol properties. Further, comparing with only two-dimensional AERONET, MISR, and/or MODIS data is a lost opportunity for important insights for model improvements. Through the analysis of aerosol properties derived from two

related, but distinctly different global climate models, we are able to provide valuable information for improving the physics of the models for future versions. Our findings can therefore be used immediately by model developers to improve aerosol treatment.

Our evaluation of model data with all available AERONET data shows the value of a multi-parameter analysis. For example, while CM3 poorly simulates seasonal AOD in Belsk and Alta Floresta ($r^2$ = 0.15 and 0.07, respectively), the seasonal

variation of SSA and α is well-simulated and improved from CM2.1 (SSA $r^2$ = 0.69 and 0.88; α $r^2$ = 0.94 and 0.95). This indicates that although seasonal AOD is poor in CM3, the model does in fact have a reasonable representation of the seasonal mixture of different aerosol types, suggesting that this is unlikely the source of the poor AOD seasonality. Further, parsing out the absorption vs. scattering AOD reveals insights into which species are under or overestimated. For example, in Kanpur, CM3 overestimates AOD magnitude by 50 to 100% from July through September. Separating out scattering and

absorption AOD shows that this is entirely due to scattering aerosols, as the absorption AOD magnitudes are consistent with observations.

The value of employing spatially collocated instruments is also shown in our study, as CALIOP revealed that AERONET comparisons can be misleading. For example, Oklahoma is the only site we looked at where AOD seasonality was better reproduced by CM3 than CM2.1 ($r^2$ = 0.97 and 0.70, respectively). This is enlightening because Oklahoma represents regions with "background" pollution – as opposed to all other sites that are extremely polluted – and therefore suggests an

5 improvement in CM3. However, comparing AERONET and model data with CALIOP reveals that the seasonal vertical distribution is better represented by CM2.1, with aerosols reaching higher elevations during peak activity; in CM3, aerosols are inaccurately constrained to the surface. Taiwan is another example where important aerosol characteristics are revealed by CALIOP; while AERONET suggests a double peak in spring and fall of similar magnitudes, the vertical structures of these peaks are extremely different, which is important for climate impacts. However, both models show similar vertical

structures during the two peak seasons. Because the vertical distributions of aerosols govern climate responses (Ocko et al., 2012; Ocko et al., 2014), model performance of vertical extinction is critical. The difference in vertical distributions also provides insight into the origins of the aerosol particles in the atmosphere. For instance, dust sources originating from northern Asia may be transported at higher elevations in the atmosphere, whereas local pollution is generally constrained closer to the surface. CALIOP further reveals the efficacy of the biomass burning injection height parameterization included

in CM3 but not CM2.1, and shows that it is not sufficient.

The comparisons of CM2.1 and CM3 aerosol properties to different observational datasets also highlight model radiative forcing biases. It is evident that for almost all biomass burning regions, the models underpredict AOD and the vertical extent of aerosols in the atmosphere; we therefore expect that the radiative forcing by carbonaceous aerosols in these regions are also underestimated, and the positive radiative forcing from black carbon is biased too low due to (i) less overall mass of

20 black carbon, and (ii) lack of black carbon forcing amplification from being located above clouds (e.g. Ocko et al. 2012).

AOD (and particularly scattering AOD) over industrialized areas provides a mixed picture in terms of biases; some sites have accurate reconstructions by the models, and others are largely overestimated (e.g. Belsk in CM2.1) or underestimated (e.g. Kanpur in CM2.1). Therefore, there may be compensating AOD biases over the globe that lead to the canceling out of associated biases in the sulfate radiative forcing over large domains, but the biases may affect the regional distribution of

25 forcing. On the other hand, the models do a fairly reasonable simulation of the seasonality (though this is not the case for CM3), single-scattering albedo, and particle size in all seven locations, leading to few biases in these properties. Because CM3 meteorology was interactive with aerosols, as opposed to CM2.1 where aerosol distributions were computed offline, CM3 needs more analysis into the dynamical feedbacks that generate aerosol seasonality from emissions data that lack seasonality. Further, while some of the discrepancy between CM2.1 and CM3 is due to different meteorology (Donner et al.,

2011), differences between model and observations also arise because the climate models are unable to reproduce specific synoptic events. Future research directions based on our findings include running model simulations that isolate aerosol changes from meteorology, emissions, and physics.

Model biases in AOD may also perturb the interhemispheric forcing asymmetry, which directly impacts climate (Ocko et al., 2014). For example, if black carbon AOD in biomass burning regions (many of which are located in the Southern Hemisphere in Africa and South America) are largely underestimated, we would expect the radiative forcing to also be underestimated in these areas. Accounting for this underestimate would yield a higher black carbon radiative forcing in the

Southern Hemisphere, and therefore less interhemispheric forcing asymmetry during peak biomass burning season. Because the majority of sulfate is located in the Northern Hemisphere, we do not expect a similar bias in the sulfate interhemispheric forcing asymmetry, which is more certain and already more pronounced than that of black carbon.

Model biases in organic carbon must also be considered. The formation of secondary organics is poorly understood, and emissions databases of carbonaceous material from non-fossil fuel combustion are limited (e.g., biofuel combustion,

cowdung burning, tea leaves burning). Organic carbon concentrations are likely too low in the models in biomass and biofuel burning regions. If the scattering by organic carbon is dominant, the underestimate may cancel out some of the bias in the underpredicted black carbon forcing in these regions, thereby retaining the stark interhemispheric forcing asymmetry exhibited by black carbon. If, on the other hand, the absorption by organic carbon is dominant, correcting this bias would further amplify the positive forcing in these areas, leading to an additional reduction in the interhemispheric forcing

asymmetry.

Overall, the model biases revealed by comparisons of model data to collocated observations may affect the interhemispheric aerosol forcing asymmetry, regional magnitudes of the forcings, and the seasonality of the forcings. Future research therefore includes quantifying how model biases translate into radiative forcing uncertainties. However, when comparing model optical properties with measurements, it is also important to account for uncertainties in the aerosol optical properties

derived from instruments, discussed in Sect. 2.

Only recently tri-dimensional compositions of aerosols are being retrieved from sun photometers and LIDAR measurements (Chaikovsky et al., 2016); algorithms have been developed to tease out the individual aerosol components from datasets produced by AERONET, MPLNET, and CALIOP (Ganguly et al., 2009a, 2009b). This is especially useful for model validation of specific aerosol components, such as strong scatterers and absorbers – sulfate and black carbon – that

significantly alter the Earth's radiation budget from human activity and are an important, albeit uncertain, aspect of climate modeling.

In conclusion, we find that considering multiple parameters and spatially collocated instruments is necessary for evaluating model performance of aerosol properties, and especially useful for determining how to improve model biases; a multi-parameter evaluation determines model strengths and weaknesses, and data from spatially collocated instruments that

provide three-dimensional compositions can reveal underlying aerosol-governing physics that are otherwise masked by

integrated column properties. We therefore recommend that future aerosol modeling studies make use of all available data (parameters and instruments) when evaluating model performance.

## Acknowledgements

We would like to thank the NOAA GFDL CM2.1 and CM3 model developers, and especially Larry Horowitz for running the CM3 simulations. We also thank the entire Cloud-Aerosol Lidar and Infrared Pathfinder Satellite Observation (CALIPSO) science team for providing CALIOP data, the principal investigators and site managers of the seven AEROET stations that we acquired data from, the NASA Langley Research Center Atmospheric Science Data Center for the monthly Level 3 MISR aerosol optical depth data, and the NASA Atmosphere Archive and Distribution System (LAADS) Distributed Active Archive Center (DAAC) for the monthly Level 3 MODIS aerosol optical depth data. We are grateful to Vaishali Naik and two anonymous reviewers for providing excellent suggestions that considerably strengthened the manuscript.

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

| | | CM2.1 | CM3 |
|---|---|---|---|
| **Total Aerosol** | Aerosol Optical Depth | 0.17 | 0.16 |
| | Absorption Optical Depth | 0.01 | 0.008 |
| | Scattering Optical Depth | 0.16 | 0.15 |
| **Black Carbon** | Emissions (TgBC yr$^{-1}$) | 11 | 8.2 |
| | Burden ($\mu$g m$^{-2}$) | 550 | 270 |
| | Aerosol Optical Depth | 0.008 | 0.004 |
| | Absorption Optical Depth | 0.006 | 0.0009 |
| | Scattering Optical Depth | 0.002 | 0.003 |
| **Sulfate** | Emissions (TgSO$_2$ yr$^{-1}$) | 147 | 108 |
| | Burden ($\mu$g m$^{-2}$) | 5000 | 3500 |
| | Aerosol Optical Depth | 0.1 | 0.07 |
| | Absorption Optical Depth | 0 | 0.004 |
| | Scattering Optical Depth | 0.1 | 0.06 |
| **Organic Carbon** | Emissions (TgC yr$^{-1}$) | 52 | 75 |
| | Burden ($\mu$g m$^{-2}$) | 2700 | 3600 |
| | Aerosol Optical Depth | 0.01 | 0.03 |
| | Absorption Optical Depth | 0 | 0.008 |
| | Scattering Optical Depth | 0.01 | 0.03 |
| **Dust** | Emissions (Tg yr$^{-1}$) | 1960 | 1221 |
| | Burden ($\mu$g m$^{-2}$) | 44000 | 27000 |
| | Aerosol Optical Depth | 0.03 | 0.018 |
| | Absorption Optical Depth | 0.005 | 0.002 |
| | Scattering Optical Depth | 0.02 | 0.016 |
| **Sea Salt** | Emissions (Tg yr$^{-1}$) | NA[a] | 6188 |
| | Burden ($\mu$g m$^{-2}$) | 9800 | 12800 |
| | Aerosol Optical Depth | 0.02 | 0.04 |
| | Absorption Optical Depth | 0 | 0 |
| | Scattering Optical Depth | 0.02 | 0.04 |

**Table 1: Global-mean present-day aerosol properties as simulated by CM2.1 (1996-2000) and CM3 (2000-2004). Emissions data for year 2000. (a) See text for details.**

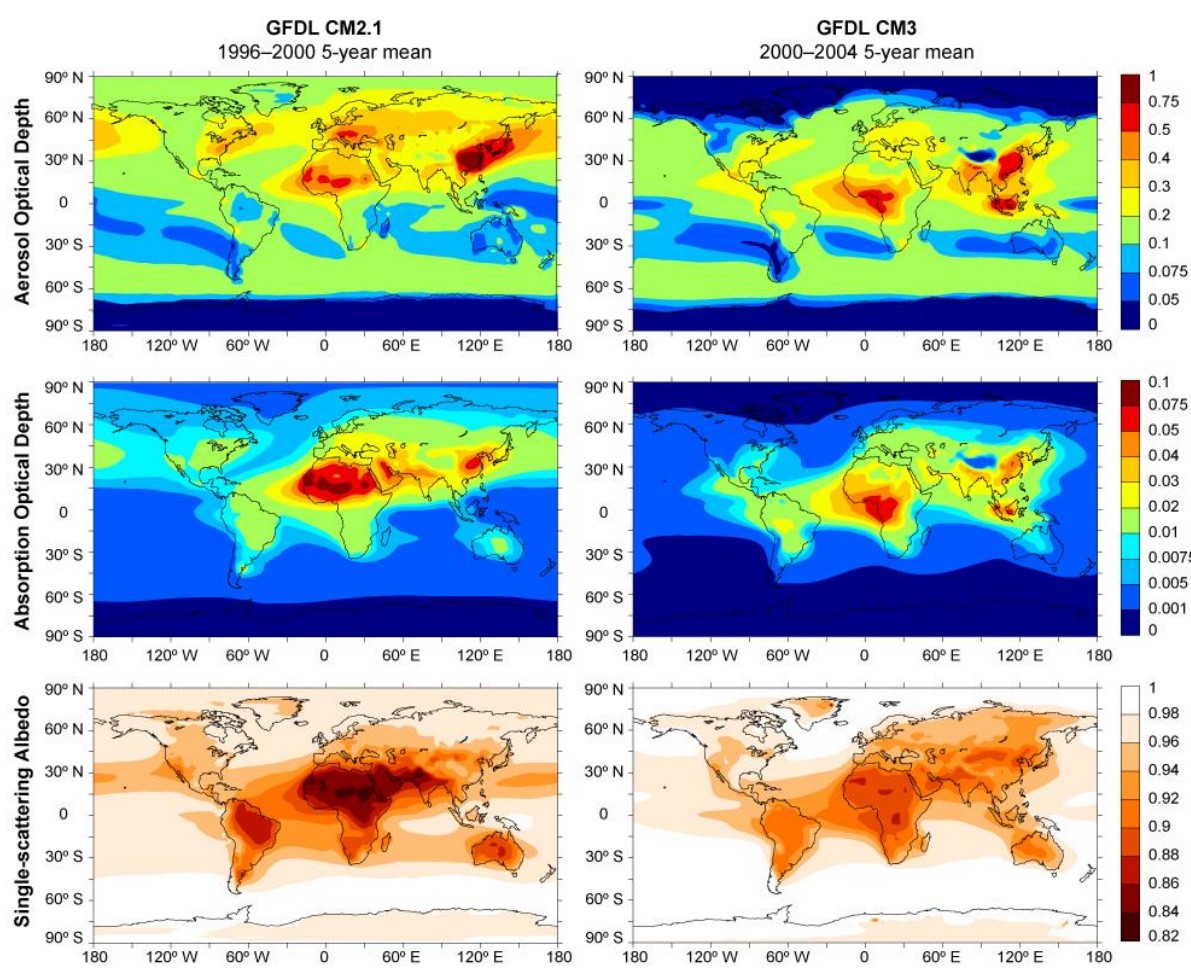

**Figure 1: Model-derived total aerosol optical properties. Five-year annual means from a 5-member historical simulation ensemble, CM2.1 present-day from 1996–2000, CM3 present-day from 2000–2004.**

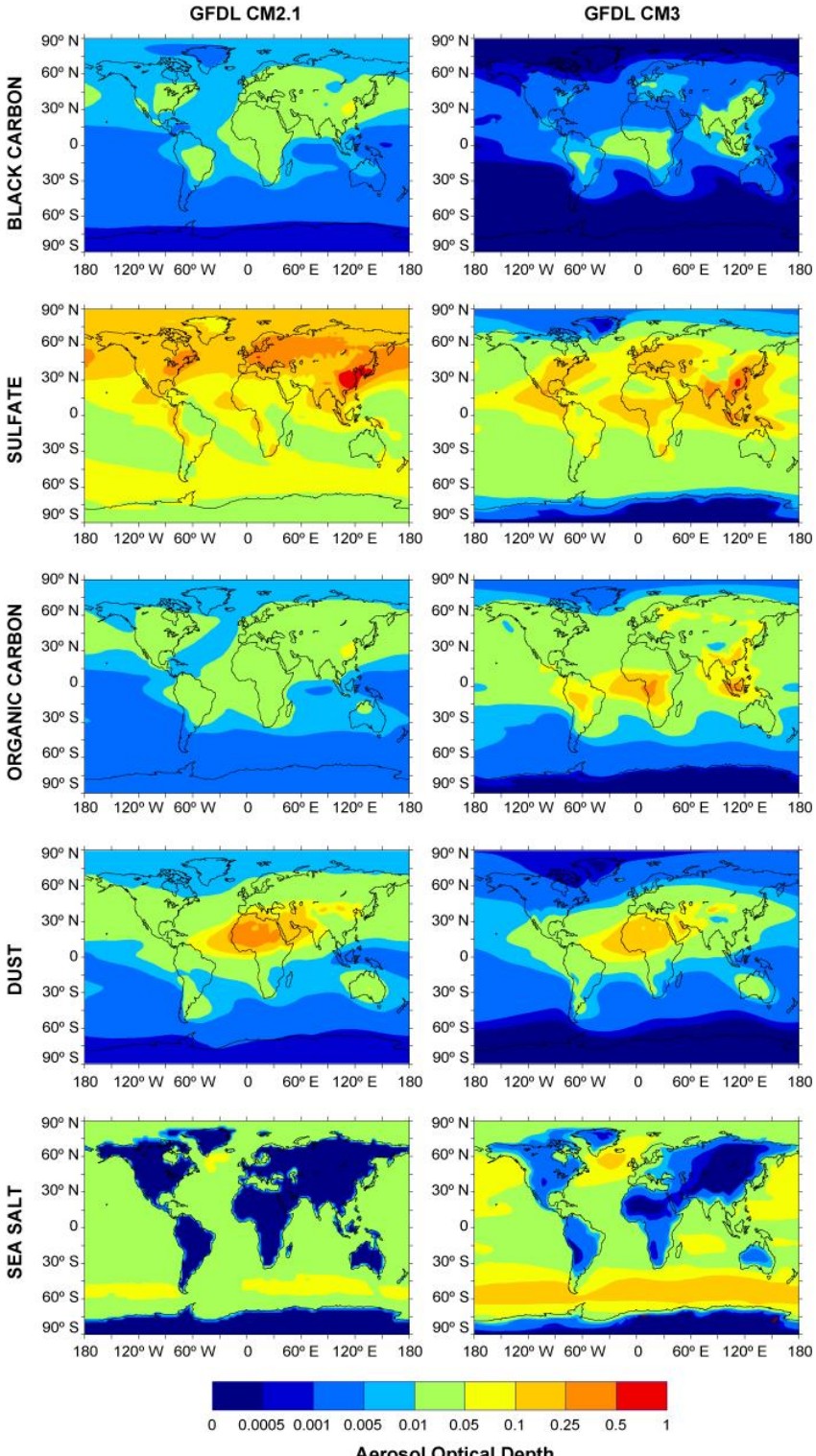

**Figure 2: Model-derived aerosol optical depth by component. Five-year annual means from a 5-member historical simulation ensemble, CM2.1 present-day from 1996–2000, CM3 present-day from 2000–2004.**

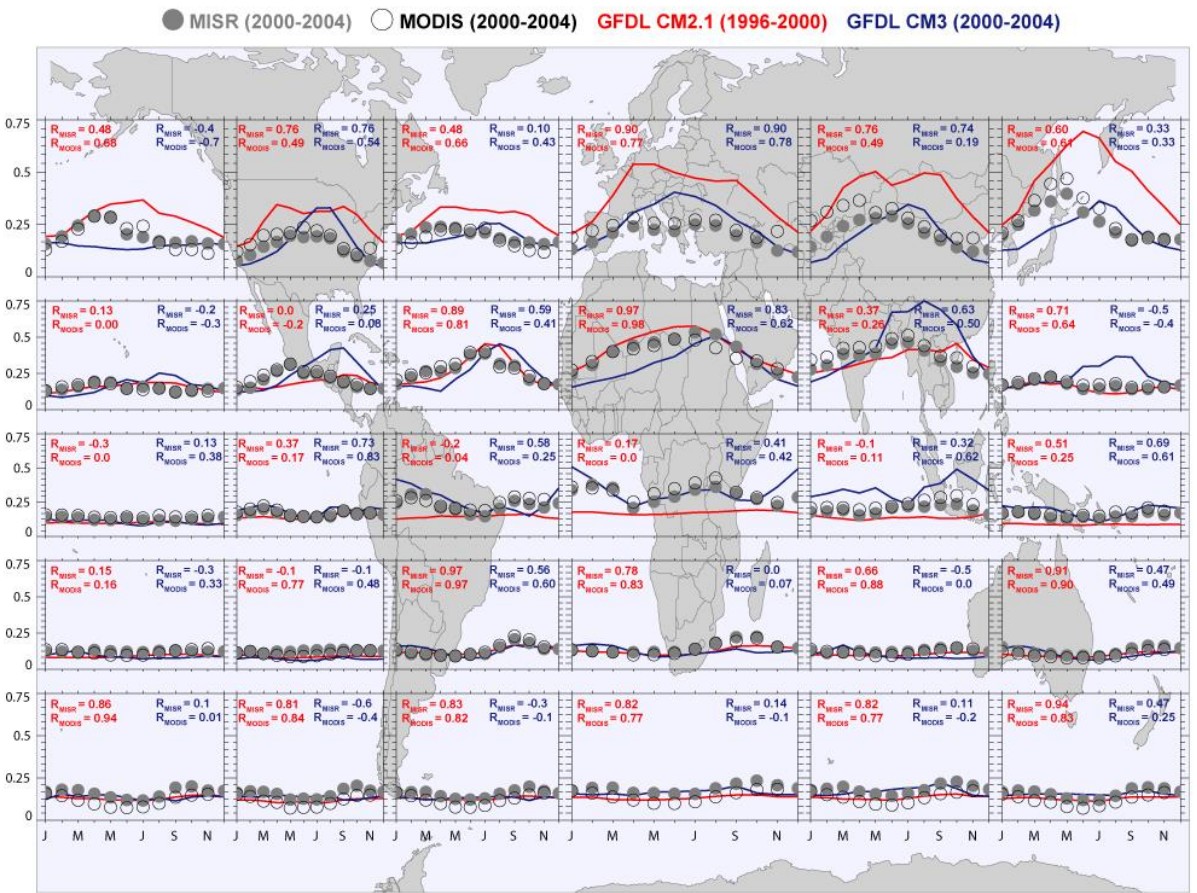

**Figure 3: Regional observed (MISR: grey circles, MODIS: open circles) and model-derived (CM2.1: red lines, CM3: blue lines) monthly aerosol optical depth at 550 nm. Values are surface-weighted averaged within each panels. Correlations between observed and simulated AOD are shown inset.**

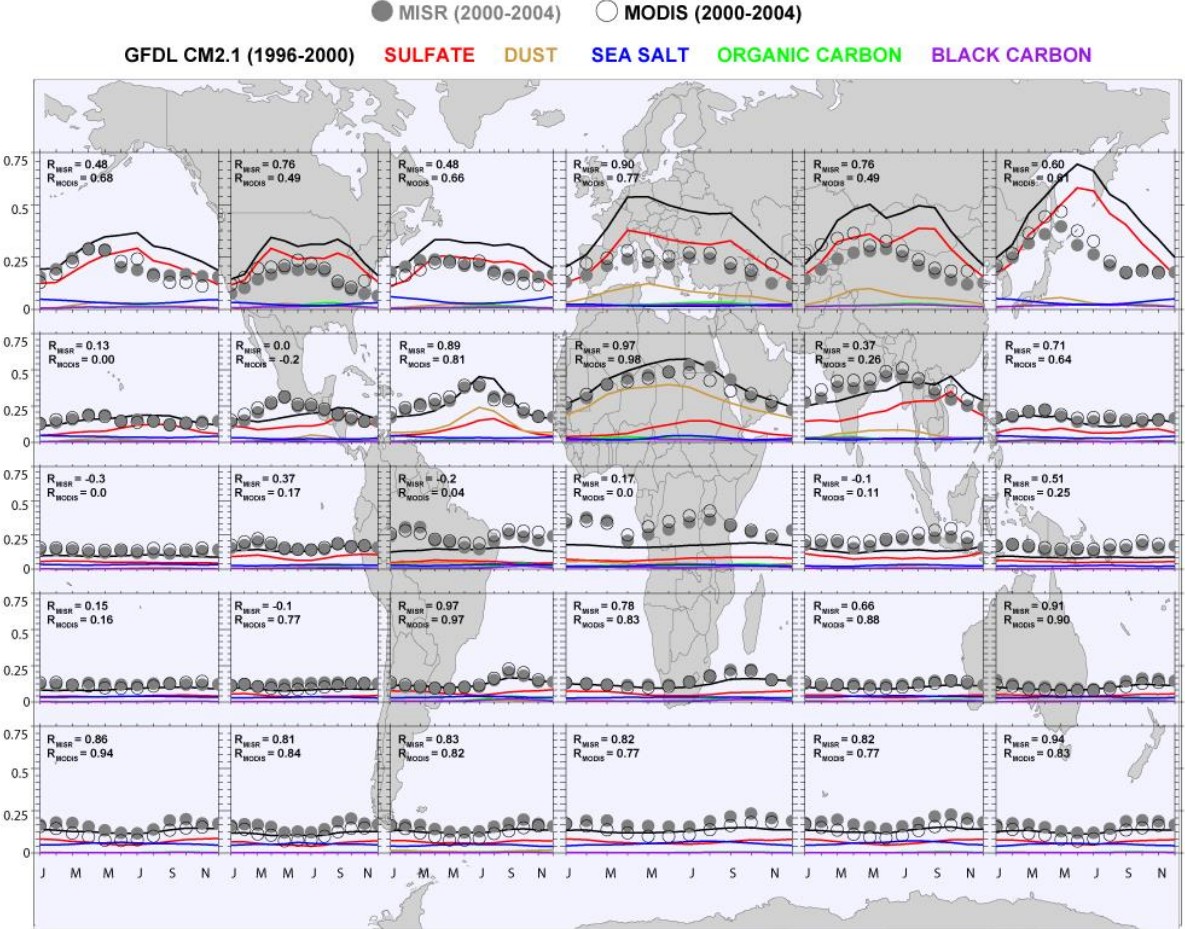

**Figure 4: Regional observed and model-derived monthly aerosol optical depth at 550 nm. Total CM2.1 AOD (black lines), individual aerosol AOD (sulfate: red, organic carbon: green, black carbon: violet, sea salt: blue, dust: brown) shown as well. Values are surface-weighted averaged within each panel. Correlations between observed and total simulated AOD are shown inset.**

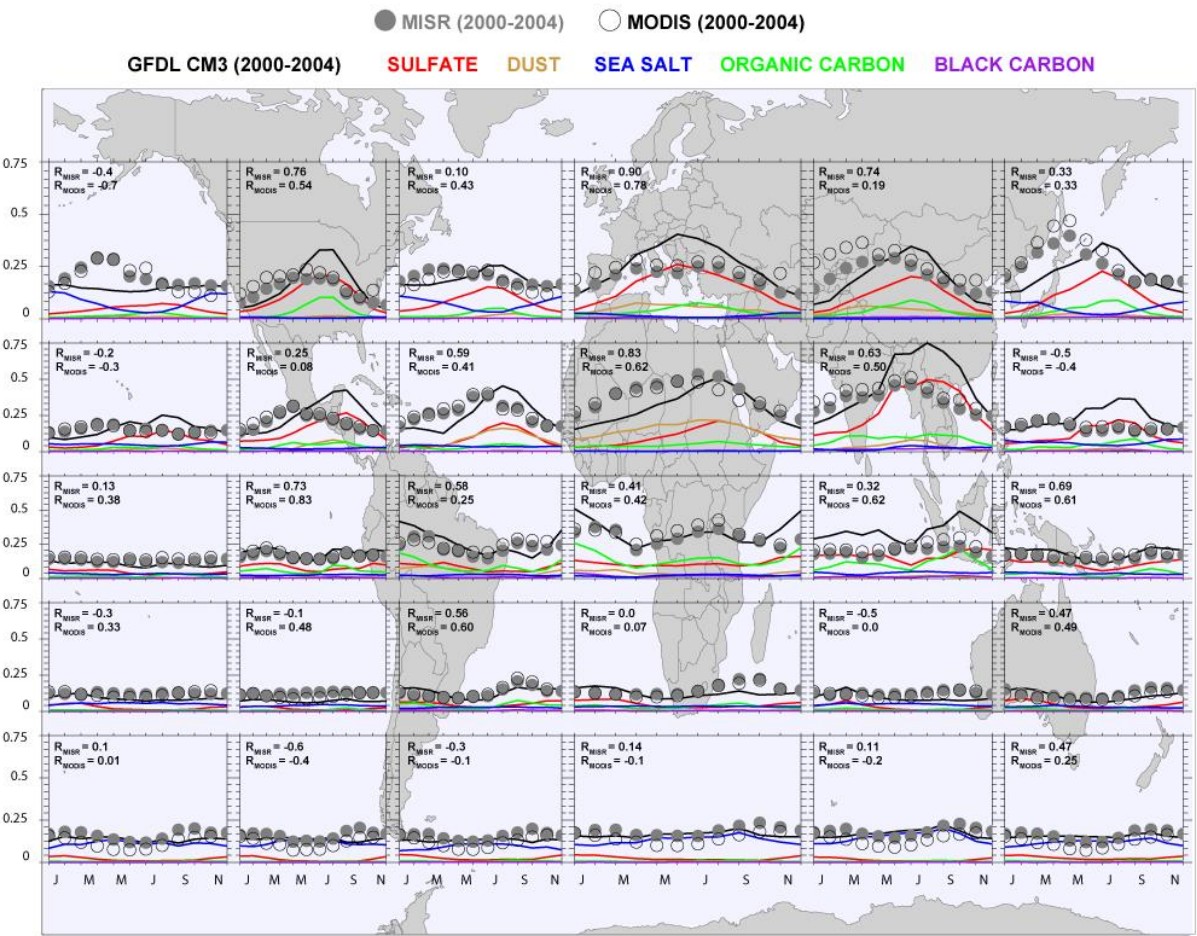

**Figure 5: Regional observed and model-derived monthly aerosol optical depth at 550 nm. Total CM2.1 AOD (black lines), individual aerosol (sulfate: red, organic carbon: green, black carbon: violet, sea salt: blue, dust: brown) AOD shown as well. Values are surface-weighted averaged within each panel. Correlations between observed and total simulated AOD are shown inset.**

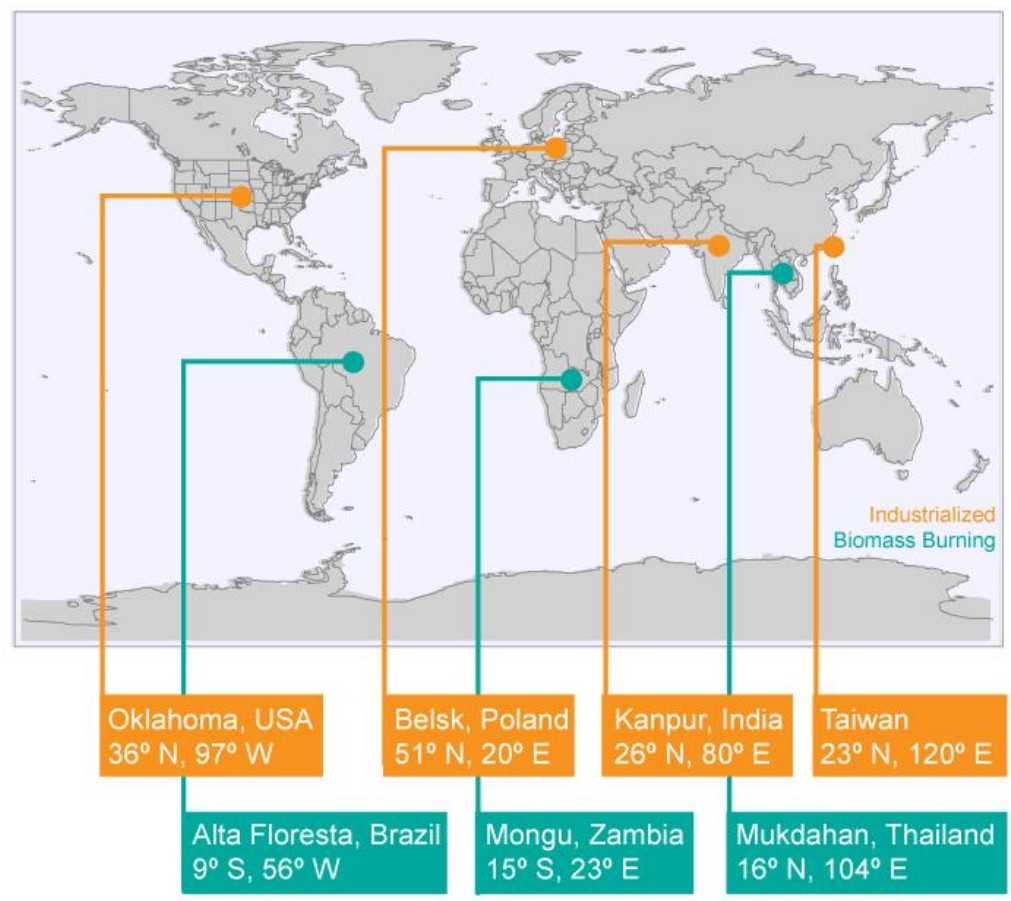

**Figure 6: Map showing the locations of comparison sites.**

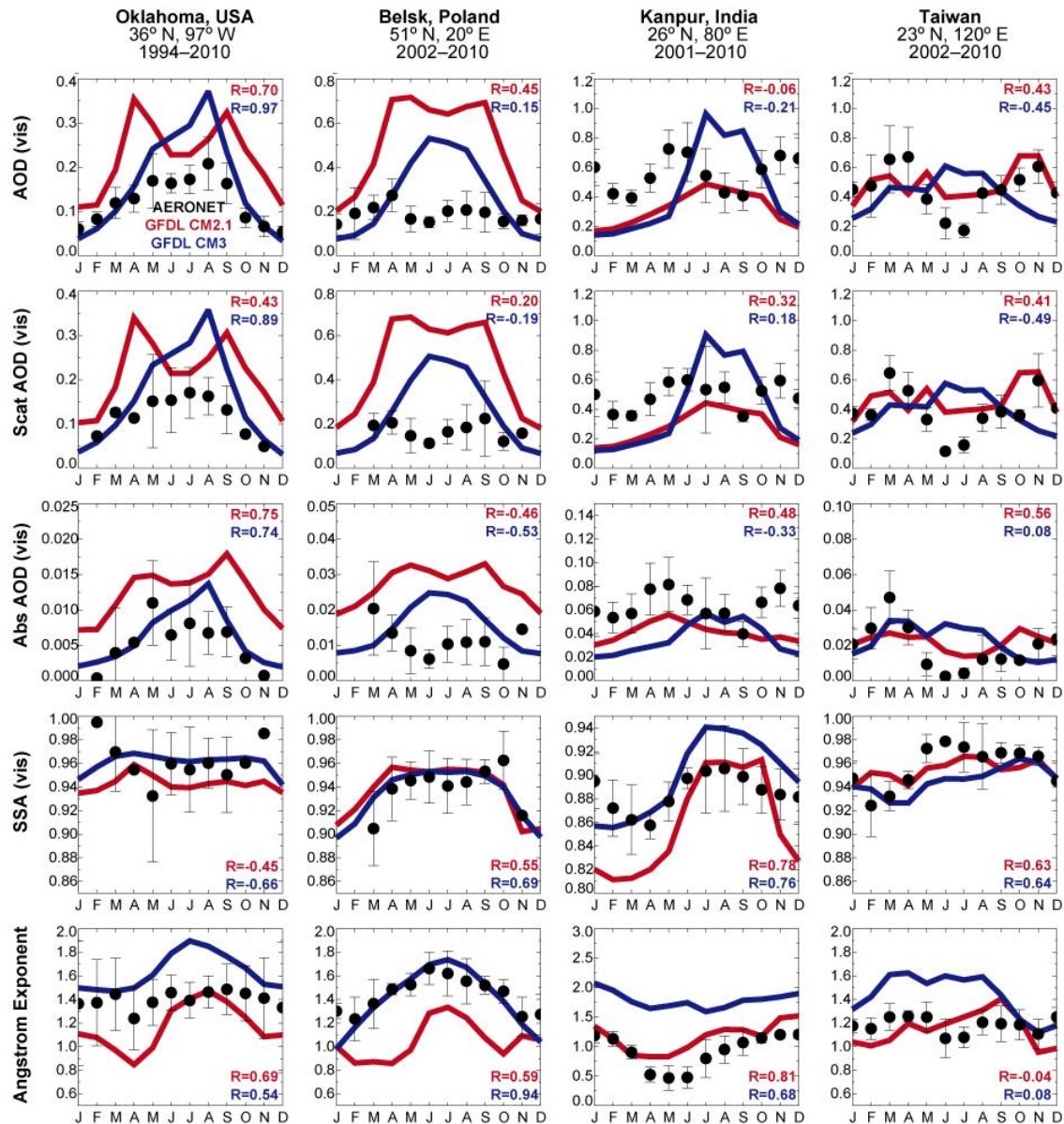

**Figure 7: Optical properties from AERONET and the models for industrialized region sites. AERONET optical properties (circles) measured at 440 nm but derived at 550 nm (see text for details), model optical properties (CM2.1: red lines, CM3: blue lines) calculated at 550 nm. Error bars represent year-to-year variability of AERONET data.**

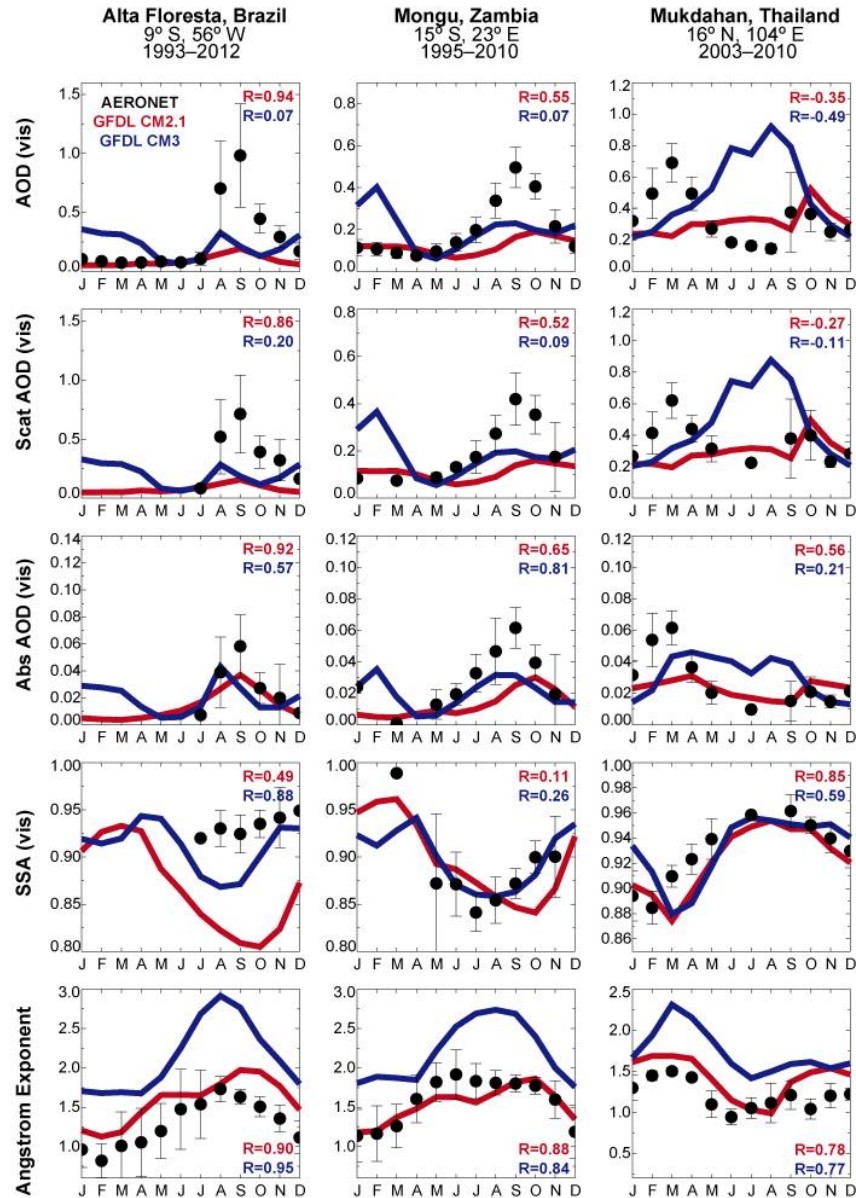

**Figure 8: Optical properties from AERONET and the models for biomass burning region sites. AERONET optical properties (circles) measured at 440 nm but derived at 550 nm (see text for details), model optical properties (CM2.1: red lines, CM3: blue lines) calculated at 550 nm. Error bars represent year-to-year variability of AERONET data.**

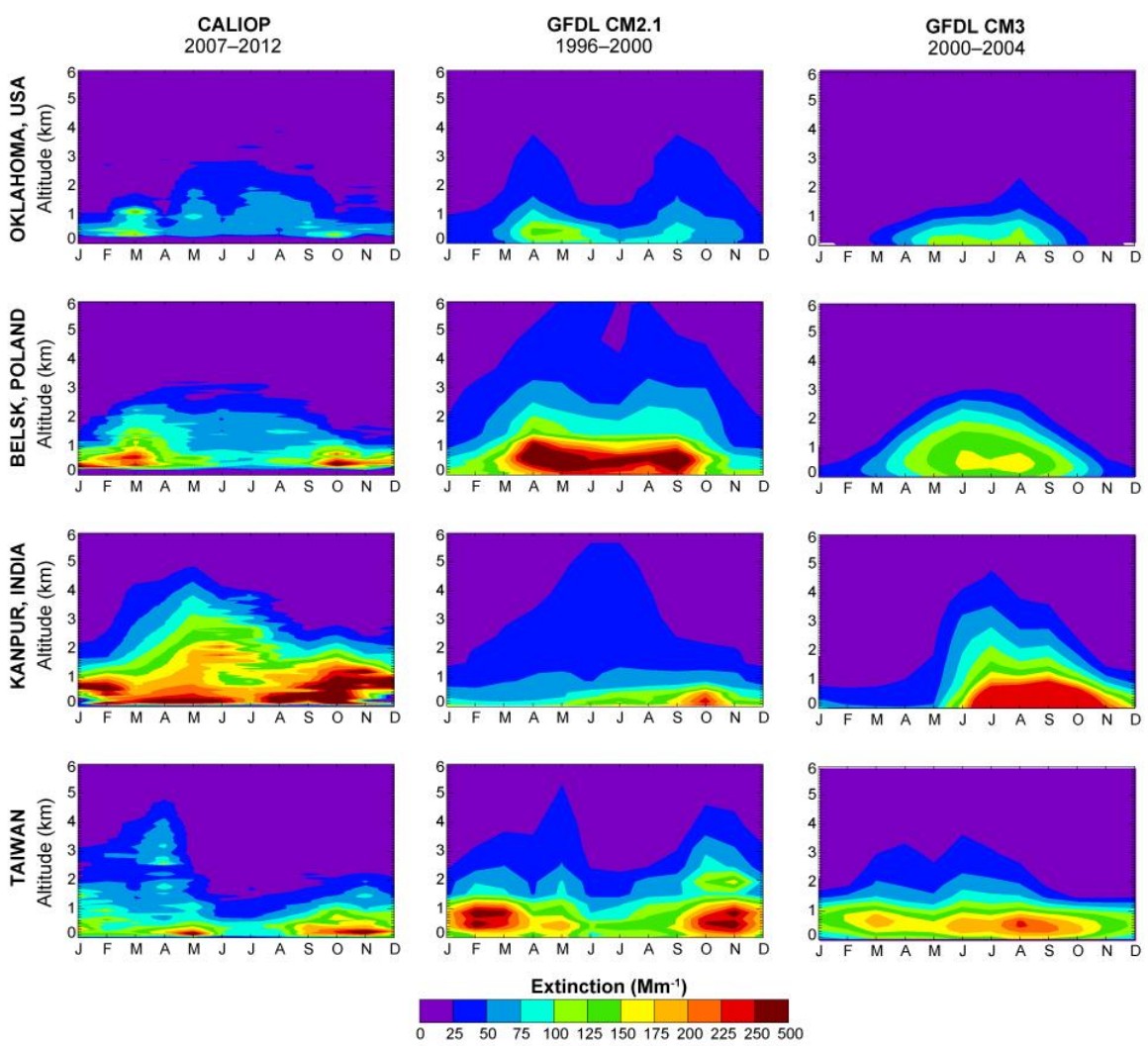

**Figure 9: Extinction coefficients (Mm$^{-1}$) from CALIOP (left panels) and the models (CM2.1: middle panels, CM3: right panels) for four industrialized sites. CALIOP extinctions measured at 550 nm, model parameters calculated at 550 nm.**

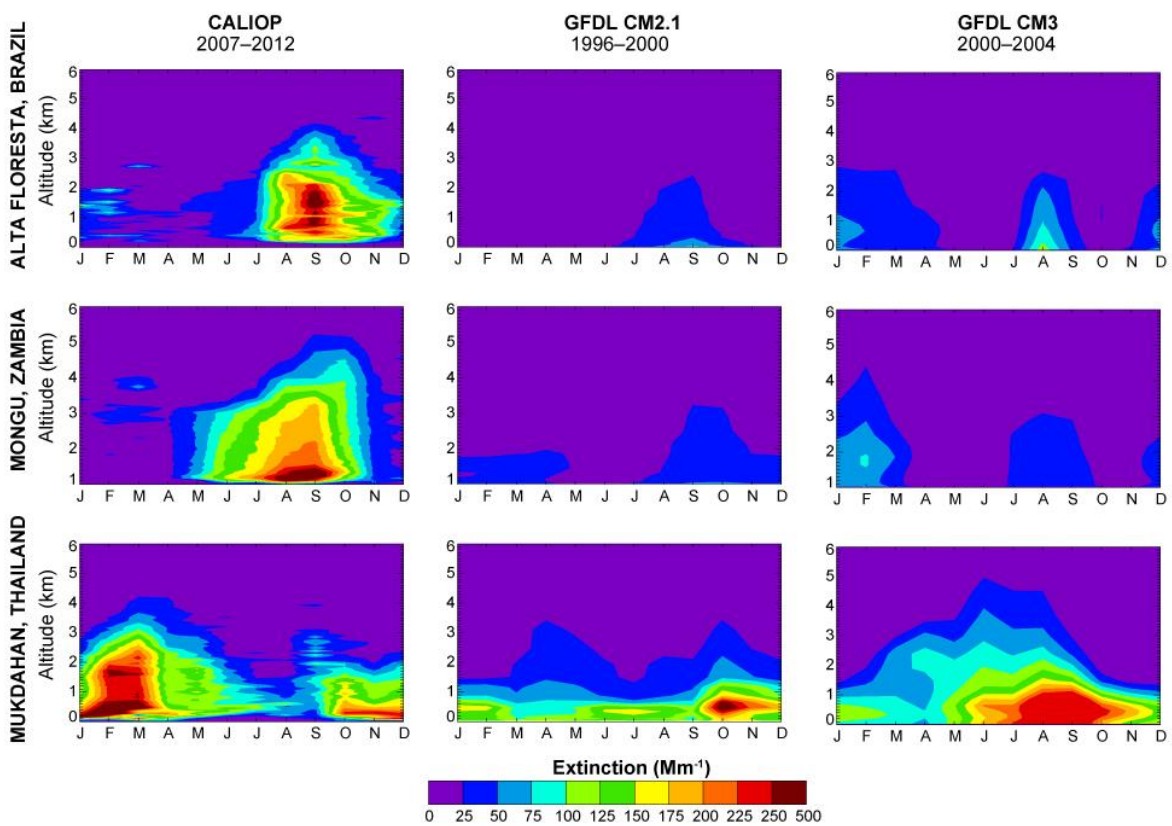

**Figure 10: Extinction coefficients (Mm⁻¹) from CALIOP and the models for three biomass burning sites. CALIOP extinctions measured at 550 nm, model (CM2.1: middle panels, CM3: right panels) parameters calculated at 550 nm.**