# Peer review of "Comparing multiple model-derived aerosol optical properties to spatially collocated ground-based and satellite measurements"

_Atmospheric Chemistry and Physics, 2016_

## Short Comment (SC1) · 21 Oct 2016

This is a nice study of various aspects of modelled aerosol that may be measured remotely. Such measurements however tend to be sparse and this introduces sampling issues. Although the authors say they use collocated observations, I found no explanation of their methodology. Possibly, this collocation is purely spatial? However, temporal collocation is known to have a big impact as well: http://www.atmos-chemphys.net/16/1065/2016/

For instance, the CALIOP dataset consists of monthly averages, but most locations will

only be visited a few times during a month. Also, absorptive AOT can only be reliably measured by AERONET when AOT is high. In both cases, comparing normal monthly model averages to the observations would introduce sampling artefacts.

Maybe the authors can provide a bit more detail on how they dealt with such problems?

regards, Nick Schutgens

---

## Referee Comment (RC1) · Anonymous Referee #2 · 14 Nov 2016

In this manuscript, "Comparing multiple model-derived aerosol optical properties to collocated ground-based and satellite measurements" the authors compare two different versions of the NOAA GFDL model with measurements of aerosol optical properties. They demonstrate the importance of looking at more than just the AOD when assessing the model performance and highlight deficiencies in the model representation of aerosol, such as biomass burning aerosol not lofted high enough in either model. The research clearly highlights the difficulties in modeling basic aerosol seasonality and loading in polluted regions. However, most AEROCOM studies do look at more than just the AOD when assessing the aerosol in models (e.g. Kinne et al., 2006, Huneeus

et al., 2011). Therefore, I'm not sure how novel the multiple-metric approach truly is, a point that is highlighted in the abstract and throughout the work. The research presented is valuable but some aspects of the research need revisiting and the conclusions need improving.

Major Comments

1) I'm left feeling that the model representation of aerosols is generally poor in the regions compared, and that this might be a combination of emissions (definitely for biomass burning), spatial resolution, potentially optical properties, aerosol size distribution etc. While the authors show that comparing multiple metrics with observations can provide more insight, there is little in the way of concrete evidence that the those insights have helped improve the understanding of the discrepancies between model and observations. I don't mean to be overly critical, and realize the simulations are time consuming, but I think the authors must justify their choice to stop at the point of speculation and not perform further simulations to try understand which of the many plausible causes actually contribute to the observed discrepancy. Key findings should be presented more concisely if possible, and more from the viewpoint of the underlying causes rather than the models being X% higher or Y% lower than the observations which is of limited use to the reader.

2) If I understand correctly, the AERONET observations used are for 440nm whereas the model is at 550nm. This will cause a general high bias in the AERONET AOD relative to the models. The difference may be small where coarse aerosol dominates but this will increase up to maybe ∼25% in regions with fresh, fine aerosol, such as biomass burning regions. I don't think the current comparison is rigorous and recommend converting AERONET AOD to 550nm. AERONET provides AOD at multiple wavelengths (and the Angstrom Exponent) so it is trivial to calculate the AERONET AOD at 550nm.

3) Also regarding the comparison with AERONET, is the comparison of the closest

grid box to the AERONET site, or has the model grid been interpolated to the exact site location? Lack of interpolation may make a substantial difference where there are strong gradients in aerosol.

4) With the CM3 model, it is difficult to understand how much of the discrepancy with observations might arise from the climate model meteorology (rather than using re-analysis fields). The authors do average over a 5-year period using the model, but it would be useful to see the interannual variability of the models on Figure 4 & 5 and some understanding of the interannual variability in the CALIOP observations.

5) It would be interesting to use the difference between the model and the observations to understand how the error in the models translates into uncertainties in the radiative effects and the interhemispheric forcing asymmetry. These are discussed qualitatively, but is it possible to expand this into some quantitative assessment using other model output fields ( surface and TOA radiative effect, etc.)?

6) I do not think the bullet-point conclusion format works well when the results are not concise. Splitting some of the conclusions into bullet points while others remain in paragraph form sees arbitrary. Please consider revising the fragmented conclusions into a more holistic discussion of the findings and how future research should proceed based on these findings.

Minor Comments

pg1 ln 29 Aerosol can travel 1000s of km in a week, so I wouldn't say it is localized around sources. Perhaps more localized than GHGs.

pg4 ln 9 Include a reference for the optical properties of BC and dust discussed.

pg5 ln 14 Add "(see Section 3.1)" regarding "computed offline" to let the reader know this will be explained.

pg5 ln 29 Remove extra period.

pg 8 please add to the description how SOA formation is treated. This is simplified and often underestimated in many models so is a potential source of discrepancy between the observations and the models.

pg 9 ln 8 Make it clear to the reader why using different years is not expected to be an issue.

pg11 ln 31 "have better magnitudes" - please rephrase.

pg12 ln 9 Remove extra punctuation

pg18 ln7 "Very nice job", please reword.

pg19 ln29 "poor emissions databases" this is very vague. Are any of the examples given included or not?

Figures 4 & 5

-in the caption, please state what the error bars represent.

-I may have missed it in the text, but the reason for missing data at Alta Floresta and other sites should be stated. I assume it is the lack of high enough AOD during that season for SSA retrieval?

-is it possible to add CALIOP AOD to these? This would be helpful when AERONET and CALIOP are often compared qualitatively in the text.

---

## Referee Comment (RC2) · Anonymous Referee #1 · 21 Dec 2016

The manuscript by Ocko and Ginoux presents a comparative study of two versions of the GFDL model, an older (CM2.1) and a newer one (CM3), against optical properties data from AERONET and CALIOP. The manuscript is clearly written and of interest to the science community, especially those using any version of the GFDL model. The analysis focuses on 4 urban locations and 3 sites influenced by significant biomass burning. Those sites, although spread around the globe, are not representative of the global atmosphere, since they represent a very small fraction of the surface of the Earth with exceptionally high pollution levels, at least seasonally. In addition, the coarse model resolution is not capable of resolving the very localized heavy pollution of the

urban centers studied, which can lead to spurious conclusions. Although I understand that there is value in comparing a global model with urban data and the authors made a considerable effort to justify that, I firmly believe that the absence of comparisons against places where the model has a chance to give good results is critical in assessing model performance. The apparent incapability of the model to resolve urban pollution also greatly degrades model skill, ending up with a not so flattering model performance, even the newer version of it, despite the great amount of work invested over the years, which resulted in large improvements in the parameterizations since the older version. I do not recommend publication in the present form, at least not until some analysis is included from locations where there is either regional pollution or cleaner conditions.

page 1, line 14 is mentioned below as 1.14, etc.

General comments

Section 3.1 (about the older model description) has some very strong assumptions about aerosol modeling. These include the absence of nitrate (6.1), the concentrations (not fluxes) of sea salt that scales with wind speed over the ocean (6.22) (what happens over land?), the zero sea salt over 850hPa (6.23), the offline aerosols coming from different (thus inconsistent) sources (6.31-7.1), the fixed 80% RH for optical calculations which is not even used for BC and OC (7.4-5). I understand that this is an older generation model that is probably not used any more, but in any case with such assumptions the correlations with measurements is expected to be poor. The fact that the new model performance is not greatly better is very surprising. I believe that the authors made the choice of using and presenting that old model to contrast the improvements in the newer model, something very useful for both the users of the GFDL model and its output (so they will look at both model versions) but also for the people that only care about the current model skill (that will look only the newer version comparisons). However, especially for the audience that belongs to the first group, the model performance probably degrades, as presented here (e.g. Figures 4-5, 15.8-9,

and 19.16). This comparison though is biased towards the urban stations where the models are not expected to perform well, which is something that even the authors acknowledge (11.29-30). A fair comparison really needs background (not necessarily clean) stations. A great example for this is Oklahoma (10.22-11.6 and figure 4), which is the only urban station captured. This is not a surprise, since the station is not in a city, but downwind of one, and represents regional pollution.

Another argument against comparing with background and even remote stations can be found when comparing the results of Naik et al. (2013), presented in 8.30-32: The global AOD biases are within 5% or 2%, while the differences presented here are significantly larger, and frequently exceed a factor of 2 (section 4.2.1). I understand the motive to accurately capture the very high pollution regions where aerosol-climate interactions maximize, but these are not representative of the global atmosphere and should not be used as a metric of model skill, as is done here.

The discussion is overly qualitative at times, in too many places to be able to enumerate. There are several examples, most of which include wording like "slight", "reasonably", "somewhat", "a better/worse/nice job", "better magnitudes", "fairly well", "correlates well", etc. More quantitative statements need to be used throughout.

Specific comments

1.14: please put the names of the models in the abstract.

1.24-27: Longwave aerosol absorption is also an important climate driver.

3.5: …treatments IN THE TWO MODELS are…

3.11: Delete first instance of word "instruments".

3.11: Describe a bit more the cities, e.g. population, including any other information that might be useful for the reader. Throughout the manuscript there are scattered information, e.g. types of fuels burned in the area, meteorological conditions, etc. This is a good place to have them all together.

4.6-8: BC has an Ångström exponent of 1 across the visible spectrum when externally mixed (see paragraph 112 in Bond et al., 2013), while a spectral dependence is measured for coated BC aerosols. Since BC is homogeneously mixed and not coated in this study, this statement is probably misleading.

4.13: To my knowledge, hardly any model uses interpolations when doing comparisons, primarily because the model uncertainties are probably larger than the concentration gradient in a grid box. Unless the authors believe the opposite, which would then require to justify why this approach was not followed, I recommend dropping the sentence.

4.25-26: How do you use temporal colocation with CALIOP, which only has day/night profiles at specific times a day? Simply take the level 3 product and compare with the modeled monthly mean? If yes, this is not what colocation means.

5.29: delete extra dot.

7.23: . . . Second, SOME (please say which) aerosol. . .

7.25: Aerosol indirect effects are not considered in this study (5.16-17), so either drop this sentence or remind the reader.

7.27: . . .to be HOMOGENEOUSLY internally mixed. . .

7.23-27: Is there nitrate aerosol in this version of CM3? I know there is from recent publications of the same group, but is it present in this current study?

8.12: "Transportation" –> "Transport".

8.15: "property" –> "properties".

Figure 3: How do you break down the per-component AOD when internal mixing is assumed? This is important information to be in the text, e.g. in 9.21.

9.28: Why the Jaegle et al. (2011) paper is cited? Is this parameterization used in
CM3? Please say so, if yes.

Section 4.2.1 is too long. I propose splitting it in two (or three, given my request for background stations), with the second part starting 13.16.

11.7-8: Delete "Upon further investigation".

12.9: Fix typo in punctuation.

12.18: model shows –> models show.

12.30: delete both commas.

13.27: scale –> magnitude.

13.28: capture –> include.

14.3-17: Alta Floresta experienced severe deforestation at the beginning of the dataset used in the manuscript, which later declined significantly. This is probably why the error bars are too large during the dry season: not because of the strong interannual variability, but due to the steep decline of biomass burning in the area over the years. You might want to consider using a shorter period of time from the available long time series, one that is more representative of the simulated period.

14.24: shown –> present.

17.3: I might have missed it, but what is the assumption for the vertical distribution of biomass burning emissions in CM3?

18.26: properly –> accurately.

References

Bond, T. C., Doherty, S. J., Fahey, D. W., Forster, P. M., Berntsen, T., DeAngelo, B. J., Flanner, M. G., Ghan, S., Kärcher, B., Koch, D., Kinne, S., Kondo, Y., Quinn, P. K., Sarofim, M. C., Schultz, M. G., Schulz, M., Venkataraman, C., Zhang, H., Zhang, S., Bellouin, N., Guttikunda, S. K., Hopke, P. K., Jacobson, M. Z., Kaiser, J. W., Klimont, Z.,

Lohmann, U., Schwarz, J. P., Shindell, D., Storelvmo, T., Warren, S. G., and Zender, C. S.: Bounding the role of black carbon in the climate system: A scientific assessment, Journal of Geophysical Research: Atmospheres, n/a-n/a, 10.1002/jgrd.50171, 2013.

Jaegle, L., Quinn, P. K., Bates, T. S., Alexander, B., and Lin, J. T.: Global distribution of sea salt aerosols: new constraints from in situ and remote sensing observations, Atmos Chem Phys, 11, 3137-3157, DOI 10.5194/acp-11-3137-2011, 2011.

Naik, V., Horowitz, L. W., Fiore, A. M., Ginoux, P., Mao, J., Aghedo, A. M., and Levy, H.: Impact of preindustrial to present-day changes in short-lived pollutant emissions on atmospheric composition and climate forcing, Journal of Geophysical Research: Atmospheres, 118, 8086-8110, 10.1002/jgrd.50608, 2013.
* * *

---

## Author Response (AR1)

**Manuscript Ref: acp-2016-790**

**Comparing multiple model-derived aerosol optical properties to spatially collocated**

**ground-based and satellite measurements**

I. B. Ocko and P. A. Ginoux

We sincerely appreciate the thoughtful reviews of our manuscript, and thank the referees and Editor for their time. The suggestions have undoubtedly and considerably enhanced the manuscript.

Specifically, we have improved the analysis by (i) developing a new section with regional evaluation of model performance with two new instruments and three new figures, (ii) providing quantification of results throughout the text, and (iii) converting AERONET AOD to 550 nm from 440 nm. We have also refined and reformatted the conclusions, and added 12 new references.

Below, we have responded point-by-point to comments and provided information on the modifications in the text.

Responses to Interactive Short Comment #1 (N.A.J. Schutgens):

**Comment 1:** This is a nice study of various aspects of modelled aerosol that may be measured remotely. Such measurements however tend to be sparse and this introduces sampling issues. Although the authors say they use collocated observations, I found no explanation of their methodology. Possibly, this collocation is purely spatial? However, temporal collocation is known to have a big impact as well: http://www.atmos-chemphys.net/16/1065/2016/.

For instance, the CALIOP dataset consists of monthly averages, but most locations will only be visited a few times during a month. Also, absorptive AOT can only be reliably measured by AERONET when AOT is high. In both cases, comparing normal monthly model averages to the observations would introduce sampling artefacts. Maybe the authors can provide a bit more detail on how they dealt with such problems?

> **Response:** We thank the author for reading our manuscript and providing valuable feedback. The author is correct in that the collocation we are referring to is spatial. We have revised the manuscript to more clearly define this and accompanied our use of the word collocated with the word spatial, and have added text to address the temporal collocation implications as well. We thank the author for providing this reference. The added text (lines 6.9-6.11) reads: *"While the data we use from CALIOP is spatially collocated with the AERONET stations and model data, it is not temporally collocated. A recent study has shown that temporal collocation can be significant and sampling errors are introduced when it is not considered (Schutgens et al., 2016)."*
>
> We have also suggested accounting for temporal collocation as a future research direction.

**Comment 1:** In this manuscript, "Comparing multiple model-derived aerosol optical properties to collocated ground-based and satellite measurements" the authors compare two different versions of the NOAA GFDL model with measurements of aerosol optical properties. They demonstrate the importance of looking at more than just the AOD when assessing the model performance and highlight deficiencies in the model representation of aerosol, such as biomass burning aerosol not lofted high enough in either model. The research clearly highlights the difficulties in modeling basic aerosol seasonality and loading in polluted regions. However, most AEROCOM studies do look at more than just the AOD when assessing the aerosol in models (e.g. Kinne et al., 2006, Huneeus et al., 2011). Therefore, I'm not sure how novel the multiple-metric approach truly is, a point that is highlighted in the abstract and throughout the work. The research presented is valuable but some aspects of the research need revisiting and the conclusions need improving.

> **Response:** We thank the referee for their time in thoughtfully reviewing our manuscript. While AeroCom studies do look at different variables to perform regional analysis, this is different than comparing numerous aerosol parameters with data at one location. We have clarified this in the introduction (lines 2.22-2.24): *"However, most studies do not take advantage of all available datasets beyond regional analysis (Kinne et al., 2006; Huneeus et al., 2011), even though a multi-dataset approach can provide a more comprehensive picture (Miller et al., 2011)."*
>
> Our method is a step toward closing model uncertainties, where all parameters are constrained with observation. As these properties vary spatially and temporally, it is more appropriate to use spatially collocated data rather than a regional average. On the other hand, comparing at a larger scale has its own merit and we have therefore revised our manuscript by adding a new section with broad regional analysis (Section 4.2.1), three new figures (Figures 3, 4, and 5), and comparing with data from two more instruments (MODIS and MISR).

**Comment 2:** I'm left feeling that the model representation of aerosols is generally poor in the regions compared, and that this might be a combination of emissions (definitely for biomass burning), spatial resolution, potentially optical properties, aerosol size distribution etc. While the authors show that comparing multiple metrics with observations can provide more insight, there is little in the way of concrete evidence that the those insights have helped improve the understanding of the discrepancies between model and observations. I don't mean to be overly critical, and realize the simulations are time consuming, but I think the authors must justify their choice to stop at the point of speculation and not perform further simulations to try understand which of the many plausible causes actually contribute to the observed discrepancy. Key findings should be presented more concisely if possible, and more from the viewpoint of the underlying causes rather than the models being X% higher or Y% lower than the observations which is of limited use to the reader.

**Response:** We thank the referee for their feedback. To show that overall model performance is not poor, we have added a new section (Section 4.2.1) with three new figures (Figures 3, 4, and 5) to evaluate model performance on regional scales. This analysis shows that the models' representations of aerosols is satisfactory in relatively clean regions, and provides more context for why the seven locations were chosen as key regions for further evaluation.

This work is an attempt to analyze as best as possible the strengths and weaknesses of the aerosol properties which are forcing two prominent, related climate models. Similar to many climate centers, we follow precise model setup and emissions scenarios as guided by the IPCC so that model comparisons can be made. While it is not our role to test emission schemes or other climate tuning parameters, we are able to provide feedback to improve emission scenarios or aerosol properties. We have clarified this in the text (lines 2.29-2.31): *"By characterizing model strengths and weaknesses, we are able to provide feedback to improve emission scenarios and aerosol properties for future model generations."*

Further, we have presented our key findings more concisely in the conclusions section by restructuring the text, providing more meaning to the results, and adding future research directions based on our findings.

**Comment 3:** If I understand correctly, the AERONET observations used are for 440nm whereas the model is at 550nm. This will cause a general high bias in the AERONET AOD relative to the models. The difference may be small where coarse aerosol dominates but this will increase up to maybe 25% in regions with fresh, fine aerosol, such as biomass burning regions. I don't think the current comparison is rigorous and recommend converting AERONET AOD to 550nm. AERONET provides AOD at multiple wavelengths (and the Angstrom Exponent) so it is trivial to calculate the AERONET AOD at 550nm.

**Response:** We thank the referee for this suggestion, and have converted the AERONET AOD from 440nm to 550nm using the Angstrom component. Figures 6 and 7 have been modified to show the AERONET data for 550nm, and all comparisons within the text have been updated. We have also calculated the correlation coefficients between AERONET and the models for all sites and parameters to provide more quantitative assessment. Overall, converting AERONET AOD from 440 nm to 550 nm lowered total AOD in all industrial and biomass burning sites.

**Comment 4:** Also regarding the comparison with AERONET, is the comparison of the closest grid box to the AERONET site, or has the model grid been interpolated to the exact site location? Lack of interpolation may make a substantial difference where there are strong gradients in aerosol.

**Response:** The comparison between the model and AERONET is indeed the closest grid box, with no interpolation. This was written in the former Section 2.1, but we have now removed it based on feedback from another referee that says to their knowledge hardly any model interpolates grid box data when doing comparisons because the model uncertainties are often larger than the concentration gradient in the grid box. We have instead provided clarity and discussion of this in Section 3 (lines 7.5-7.9): *"Lack of interpolation of model data in polluted regions may introduce a bias in locations with strong aerosol gradients; however, interpolation is rarely employed for comparisons with observations because the model uncertainties are often larger than the concentration gradient in the grid box."*

**Comment 5:** With the CM3 model, it is difficult to understand how much of the discrepancy with observations might arise from the climate model meteorology (rather than using reanalysis fields). The authors do average over a 5-year period using the model, but it would be useful to see the interannual variability of the models on Figure 4 & 5 and some understanding of the interannual variability in the CALIOP observations.

**Response:** This is a good point, as the discrepancy may be in large part due to climate meteorology. In addition, CM2.1 and CM3 have different physics and produce different climates (cf. Donner et al., 2011). It is also important to note that these comparisons are made with climate models which are unable to reproduce specific synoptic events. We add in the text the following (lines 17.31-18.2): *"While some of the discrepancy between CM2.1 and CM3 is due to different meteorology (Donner et al., 2011), differences between model and observations also arise because the climate models are unable to reproduce specific synoptic events."*

**Comment 6:** It would be interesting to use the difference between the model and the observations to understand how the error in the models translates into uncertainties in the radiative effects and the interhemispheric forcing asymmetry. These are discussed qualitatively, but is it possible to expand this into some quantitative assessment using other model output fields ( surface and TOA radiative effect, etc.)?

**Response:** We have looked at the clear-sky downward shortwave radiation, and it is generally larger in CM3 than CM2.1 and closer to observations from the Baseline Surface Radiation Network (Donner et al., 2011). The increases in clear-sky downward shortwave radiation are due to reduced aerosol AOD in CM3. Although correlation of AOD decreases with CM3, from a climate perspective Donner et al. (2011) showed an improved agreement of CM3 simulations of downward clear-sky surface shortwave radiation, optical depths, and coalbedo with BSRN and AERONET. These improvements made the authors conclude that the direct effects of aerosols are more realistically simulated in CM3. A quantitative assessment of how model biases translate into radiative forcing uncertainties is currently beyond the scope of this paper, but an excellent idea for

a future paper, and we appreciate the suggestion. We have added this as a possible future research direction in the conclusions section.

**Comment 7:** I do not think the bullet-point conclusion format works well when the results are not concise. Splitting some of the conclusions into bullet points while others remain in paragraph form sees arbitrary. Please consider revising the fragmented conclusions into a more holistic discussion of the findings and how future research should proceed based on these findings.

> **Response:** We thank the referee for this feedback. We have greatly improved the conclusions section by restructuring the text and consolidating the results. We have also included discussion on future research based on our findings.

**Comment 8:** Minor Comments

pg1 ln 29 Aerosol can travel 1000s of km in a week, so I wouldn't say it is localized around sources. Perhaps more localized than GHGs.
pg4 ln 9 Include a reference for the optical properties of BC and dust discussed.
pg5 ln 14 Add "(see Section 3.1)" regarding "computed offline" to let the reader know this will be explained.
pg5 ln 29 Remove extra period.
pg 8 please add to the description how SOA formation is treated. This is simplified and often underestimated in many models so is a potential source of discrepancy between the observations and the models.
pg 9 ln 8 Make it clear to the reader why using different years is not expected to be an issue.
pg11 ln 31 "have better magnitudes" - please rephrase.
pg12 ln 9 Remove extra punctuation
pg18 ln7 "Very nice job", please reword.
pg19 ln29 "poor emissions databases" this is very vague. Are any of the examples given included or not?
Figures 4 & 5
-in the caption, please state what the error bars represent.
-I may have missed it in the text, but the reason for missing data at Alta Floresta and other sites should be stated. I assume it is the lack of high enough AOD during that season for SSA retrieval?
-is it possible to add CALIOP AOD to these? This would be helpful when AERONET and CALIOP are often compared qualitatively in the text.

> **Response:** We thank the referee for helping us improve the manuscript by clarifying major and minor points and tightening the text. We have made all of the above modifications. However, while we have plotted the CALIOP AOD in comparison to the models and AERONET, we have ultimately decided not to include it in the former Figs. 4 and 5 because it is known to be very problematic when the extinctions are integrated vertically, and thus may provide misleading information.

**Comment 1:** The manuscript by Ocko and Ginoux presents a comparative study of two versions of the GFDL model, an older (CM2.1) and a newer one (CM3), against optical properties data from AERONET and CALIOP. The manuscript is clearly written and of interest to the science community, especially those using any version of the GFDL model. The analysis focuses on 4 urban locations and 3 sites influenced by significant biomass burning. Those sites, although spread around the globe, are not representative of the global atmosphere, since they represent a very small fraction of the surface of the Earth with exceptionally high pollution levels, at least seasonally. In addition, the coarse model resolution is not capable of resolving the very localized heavy pollution of the urban centers studied, which can lead to spurious conclusions. Although I understand that there is value in comparing a global model with urban data and the authors made a considerable effort to justify that, I firmly believe that the absence of comparisons against places where the model has a chance to give good results is critical in assessing model performance. The apparent incapability of the model to resolve urban pollution also greatly degrades model skill, ending up with a not so flattering model performance, even the newer version of it, despite the great amount of work invested over the years, which resulted in large improvements in the parameterizations since the older version. I do not recommend publication in the present form, at least not until some analysis is included from locations where there is either regional pollution or cleaner conditions.

> **Response:** We understand the referee's concerns, and have added three new figures (Figs. 3, 4, and 5) and a new section of the paper (Section 4.2.1) to analyze model performance on a regional scale. To add to our existing evaluation of model performance with AERONET and CALIOP data, we use MODIS and MISR data to evaluate model AOD in all regions of the world, and calculate correlation coefficients to provide quantitative assessment. We show that in cleaner regions, both models successfully reproduce AOD magnitudes. In many polluted regions, there is an improvement in AOD from CM2.1 to CM3, but the seasonality performance declines. In addition to regional analysis of overall AOD from each model, we parse out the AOD by aerosols species, in order to better understand model biases. This added analysis provides context and motivation for the rest of our study, where we pinpoint and more deeply evaluate key regions where the models do not perform well. Through analysis of multiple aerosol parameters and spatially collocated instruments, we are able to better characterize model successes and failures. This will provide important information for future model improvements. We thank the referee for the suggestion to include an analysis of regional and cleaner conditions, and in doing so we have considerably enhanced the paper while providing the foundational context for the rest of our analysis.

**Comment 2:** Section 3.1 (about the older model description) has some very strong assumptions about aerosol modeling. These include the absence of nitrate (6.1), the concentrations (not fluxes) of sea salt that scales with wind speed over the ocean (6.22) (what happens over land?), the zero sea salt over 850hPa (6.23), the offline aerosols

coming from different (thus inconsistent) sources (6.31-7.1), the fixed 80% RH for optical calculations which is not even used for BC and OC (7.4-5). I understand that this is an older generation model that is probably not used any more, but in any case with such assumptions the correlations with measurements is expected to be poor. The fact that the new model performance is not greatly better is very surprising. I believe that the authors made the choice of using and presenting that old model to contrast the improvements in the newer model, something very useful for both the users of the GFDL model and its output (so they will look at both model versions) but also for the people that only care about the current model skill (that will look only the newer version comparisons). However, especially for the audience that belongs to the first group, the model performance probably degrades, as presented here (e.g. Figures 4-5, 15.8-9, and 19.16). This comparison though is biased towards the urban stations where the models are not expected to perform well, which is something that even the authors acknowledge (11.29-30). A fair comparison really needs background (not necessarily clean) stations. A great example for this is Oklahoma (10.22-11.6 and figure 4), which is the only urban station captured. This is not a surprise, since the station is not in a city, but downwind of one, and represents regional pollution.

> **Response:** The referee is correct in the assumption of why we compare both CM2.1 and CM3 with observations. To provide the larger context for the basis of our study, and to offer a fair comparison, we have included a new section (Section 4.2.1), new figures (Figs. 3, 4, and 5), and new instruments that look at background regions. While the older generation model (CM2.1) is not used much anymore, and CM3 does indeed improve AOD magnitudes in almost all regions of the globe, there is a decline in seasonal performance from CM2.1 to CM3. By further investigating key regions that are problematic in models, we are able to pinpoint model successes and failures such that future model generations can improve aerosol distributions and optics. Further, we have added text to emphasize the significance of the Oklahoma analysis (in providing a more representative characterization of model performance) as compared with other highly polluted locations (lines 13.5-13.6): *"The site in Oklahoma is in a rural environment compared to the other urban sites we have chosen for model evaluation, and therefore represents areas with background pollution."*

**Comment 3:** Another argument against comparing with background and even remote stations can be found when comparing the results of Naik et al. (2013), presented in 8.30-32: The global AOD biases are within 5% or 2%, while the differences presented here are significantly larger, and frequently exceed a factor of 2 (section 4.2.1). I understand the motive to accurately capture the very high pollution regions where aerosol-climate interactions maximize, but these are not representative of the global atmosphere and should not be used as a metric of model skill, as is done here.

> **Response:** We agree with the referee that it is important to provide a spatially broader analysis, especially as to not bias the impression of overall model skill based on a selective analysis. To represent the global atmosphere at large, and as discussed in our response to Comment 1, we have added a new section (Section

4.2.1), new figures (Figures 3, 4, and 5), and new instruments to our existing study. We show that overall CM3 improves aerosol AOD magnitudes, but seasonality deteriorates. In unpolluted regions, both models perform well.

Further, we clarify in the text that the purpose of this study is to not adjudicate overall model performance, but rather to use a specific set of tools (multiple aerosol parameters and collocated instruments) to characterize model strengths and weaknesses to aid in future improvements. The modified text reads (lines 2.27-2.31): *"Here we show that comparing multiple model-simulated aerosol properties – from two prominent, related climate models with vastly different aerosol treatments – to available datasets from spatially collocated ground-based and satellite instruments is important for determining model biases. By characterizing model strengths and weaknesses, we are able to provide feedback to improve emission scenarios and aerosol properties for future model generations."* And (lines 3.4-3.7): *"Because the aerosol treatments in the two models are starkly different, as we present in Section 3, comparing multiple optical properties with spatially collocated instruments is especially useful in identifying possible sources of error which are otherwise challenging to determine."*

**Comment 4:** The discussion is overly qualitative at times, in too many places to be able to enumerate. There are several examples, most of which include wording like "slight", "reasonably", "somewhat", "a better/worse/nice job", "better magnitudes", "fairly well", "correlates well", etc. More quantitative statements need to be used throughout.

**Response:** We have considerably increased the quantification of our analysis. We have omitted several qualitative statements, supplemented the discussion with correlation coefficients, and also provided correlation coefficients for all model and AERONET comparisons in Figures 7 and 8, as well as for model and MISR/MODIS comparisons in the new Figures 3, 4, and 5.

**Comment 5:** Specific comments
1.14: please put the names of the models in the abstract.
1.24-27: Longwave aerosol absorption is also an important climate driver.
3.5: : : :treatments IN THE TWO MODELS are: : :
3.11: Delete first instance of word "instruments".
3.11: Describe a bit more the cities, e.g. population, including any other information that might be useful for the reader. Throughout the manuscript there are scattered information, e.g. types of fuels burned in the area, meteorological conditions, etc. This is a good place to have them all together.
4.6-8: BC has an Ångström exponent of 1 across the visible spectrum when externally mixed (see paragraph 112 in Bond et al., 2013), while a spectral dependence is measured for coated BC aerosols. Since BC is homogeneously mixed and not coated in this study, this statement is probably misleading.
4.13: To my knowledge, hardly any model uses interpolations when doing comparisons, primarily because the model uncertainties are probably larger than the concentration

gradient in a grid box. Unless the authors believe the opposite, which would then require to justify why this approach was not followed, I recommend dropping the sentence.

4.25-26: How do you use temporal colocation with CALIOP, which only has day/night profiles at specific times a day? Simply take the level 3 product and compare with the modeled monthly mean? If yes, this is not what colocation means.

5.29: delete extra dot.

7.23: : : : Second, SOME (please say which) aerosol: : :

7.25: Aerosol indirect effects are not considered in this study (5.16-17), so either drop this sentence or remind the reader.

7.27: : : :to be HOMOGENEOUSLY internally mixed: : :

7.23-27: Is there nitrate aerosol in this version of CM3? I know there is from recent publications of the same group, but is it present in this current study?

8.12: "Transportation" –> "Transport".

8.15: "property" –> "properties".

Figure 3: How do you break down the per-component AOD when internal mixing is assumed? This is important information to be in the text, e.g. in 9.21.

9.28: Why the Jaegle et al. (2011) paper is cited? Is this parameterization used in CM3? Please say so, if yes.

Section 4.2.1 is too long. I propose splitting it in two (or three, given my request for background stations), with the second part starting 13.16.

11.7-8: Delete "Upon further investigation".

12.9: Fix typo in punctuation.

12.18: model shows –> models show.

12.30: delete both commas.

13.27: scale –> magnitude.

13.28: capture –> include.

14.3-17: Alta Floresta experienced severe deforestation at the beginning of the dataset used in the manuscript, which later declined significantly. This is probably why the error bars are too large during the dry season: not because of the strong interannual variability, but due to the steep decline of biomass burning in the area over the years. You might want to consider using a shorter period of time from the available long time series, one that is more representative of the simulated period.

14.24: shown –> present.

17.3: I might have missed it, but what is the assumption for the vertical distribution of biomass burning emissions in CM3?

18.26: properly –> accurately.

> **Response:** We thank the referee for the careful and thoughtful review of our manuscript. We have made the requested modifications and clarifications, which have substantially improved the manuscript.

[revised manuscript text omitted]

Datasets measured or derived from AERONET stations are useful in that they provide observations of various properties such as aerosol optical depth (AOD), aerosol absorption optical depth (AAOD), single-scattering albedo (SSA), and the Ångström exponent ($\alpha$). While studies often compare one or two parameters (e.g. Kinne et al., 2006), the availability of multiple parameters is valuable in evaluating aerosols in a model. Further, comparing AERONET column data with collocated aerosol profile observations (from CALIOP) can provide insight into the vertical structure of extinction, which is also simulated by models.

Datasets measured or derived from AERONET stations are useful in that they provide observations of various properties such as aerosol optical depth (AOD), aerosol absorption optical depth (AAOD), single scattering albedo (SSA), and the Ångström exponent ($\alpha$). While studies often compare one or two parameters (e.g. Kinne et al., 2006), the availability of

**4.2 Evaluating multiple aerosol parameters in polluted regions**

5 Comparisons of the five-year averages of model data (CM2.1 and CM3) with averages of all available AERONET data are found in Figs. 7 (polluted cities) and 8 (biomass burning regions). Aerosol parameters compared include AOD, scattering AOD, AAOD, SSA, and α. The error bars for the AERONET data represent year-to-year variability in the available data. Correlation coefficients for monthly mean model versus AERONET data is shown inset.

10 ~~Belsk, underestimates AOD in Kanpur, slightly underestimates AOD in Taiwan, and underestimates AOD in all biomass burning sites (Alta Floresta, Mongu, Mukdahan). CM3 reasonably reproduces the AOD magnitudes in Oklahoma, Belsk, Kanpur, and Mukdahan, although the seasonality of AOD (derived by humidity influence on optical depth as opposed to emissions) is weakly captured by CM3, even for biomass burning regions that do contain seasonality in their emissions inventories. Both models consistently reproduce the magnitudes and seasonality of single-scattering albedo and the~~
15

[revised manuscript text omitted]

For Taiwan is the only site where , AERONET suggests considerably higher total AOD than the vertically-integrated CALIOP data (not shown). As compared to AERONET data, the Both CM2.1 and CM3 models have magnitudes more consistent with AERONET than CALIOP, although they both correctly reproduces magnitudes of AOD and slightly underestimates the springtime maxima by a factor of 1.5. However, while the AERONET comparison suggests that CM2.1 underestimates extinction during the peak season, comparison of the vertical extinction profile However, the CM2.1 model's vertical extinction profile isshows that CM2.1 extinction is constrained at the surface and considerably larger than that of CALIOP by over a factor of four; on the other hand, CALIOP data shows that the extinction profile extends up to 4 km in elevation (similar magnitudes of extinction in CM2.1 only reach 2 km.) While the springtime AOD peak in AERONET is slightly larger than the peak in autumn, CALIOP shows large differences in the vertical distribution of extinction during spring and autumn. During springtime, aerosol extinction reaches higher elevations than during autumn. This is consistent with studies showing long-range high-elevation transport of dust (Lin et al., 2007), and also consistent with ground-based LIDAR measurements in Taiwan (Chen et al., 2009). Recall that the main springtime sources of aerosols in Taiwan (other than industry) are dust transported from the north, and nearby biomass burning. During autumn, high extinctions are constrained closer to the ground. Interestingly, the total column optical depth when computed from CALIOP data show that the overall AODs are similar for spring and autumn, even though their vertical distributions vary tremendously. This shows the value of instruments like CALIOP in their ability to resolve aerosol vertical profiles. The model, on the other hand, does

not accurately distinguish the differences in the vertical profiles over Taiwan, and springtime and autumn extinction distributions are fairly comparable.

The springtime AOD from AERONET and the extinction vertical profile from CALIOP emphasize the importance of the vertical distribution of aerosols in the atmosphere. 
[revised manuscript text omitted]

~~Compared to the different observational datasets, the CM2.1 model overestimates (<50%) aerosol optical depth (AOD) in Oklahoma, overestimates (300%) AOD in Belsk, underestimates (100%) AOD in Kanpur, underestimates (<50%) AOD in Taiwan, and considerably underestimates (by a factor of four) the peak AOD magnitude and vertical extent at all biomass burning sites (Alta Floresta, Mongu, Mukdahan). The CM3 model, with improved aerosol treatment, does a better job in reproducing optical depth/extinction magnitudes as compared to CM2.1, as found in Donner et al. (2011), but a worse job with recreating seasonality, which is reproduced reasonably by CM2.1 despite the fact that the majority of emissions are aseasonal. Both models do a very nice job reproducing single scattering albedo and the Ångström exponent, indicative of the types of aerosols present, with few exceptions.~~

Comparing multiple aerosol optical properties derived by models to measurements from collocated instruments both identifies opportunities for the improvement of modeling aerosol distributions, as well as reveals important aspects governing aerosol properties. Further, comparing with only two-dimensional AERONET, MISR, and/or MODIS data is a lost opportunity for important insights for model improvements. Through the analysis of aerosol properties derived from two related, but distinctly different global climate models, we are able to provide valuable information for improving the physics of the models for future versions.

Our evaluation of model data with all available AERONET data shows the value of a multi-parameter analysis. For example, while CM3 poorly simulates seasonal AOD in Belsk and Alta Floresta ($r^2$ = 0.15 and 0.07, respectively), the seasonal variation of SSA and α is well-simulated and improved from CM2.1 (SSA $r^2$ = 0.69 and 0.88; α $r^2$ = 0.94 and 0.95). This indicates that although seasonal AOD is poor in CM3, the model does in fact have a reasonable representation of the seasonal mixture of different aerosol types, suggesting that this is unlikely the source of the poor AOD seasonality. Further, parsing out the absorption vs. scattering AOD reveals insights into which species are under or overestimated. For example, in Kanpur, CM3 overestimates AOD magnitude by 50 to 100% from July through September. Separating out scattering and absorption AOD shows that this is entirely due to scattering aerosols, as the absorption AOD magnitudes are consistent with observations.

The value of employing spatially collocated instruments is also shown in our study, as CALIOP revealed that AERONET comparisons can be misleading. For example, Oklahoma is the only site we looked at where AOD seasonality was better reproduced by CM3 than CM2.1 ($r^2$ = 0.97 and 0.70, respectively). This is enlightening because Oklahoma represents regions with "background" pollution – as opposed to all other sites that are extremely polluted – and therefore suggests an improvement in CM3. However, comparing AERONET and model data with CALIOP reveals that the vertical distribution is better represented by CM2.1, with aerosols reaching higher elevations during peak activity; in CM3, aerosols are inaccurately constrained to the surface. Taiwan is another example where important aerosol characteristics are revealed by CALIOP; while AERONET suggests a double peak in spring and fall of similar magnitudes, the vertical structures of these peaks are extremely different, which is important for climate impacts. However, both models show similar vertical structures during the two peak seasons. Because the vertical distributions of aerosols govern climate responses (Ocko et al., 2012; Ocko et al., 2014), model performance of vertical extinction is critical. The difference in vertical distributions also provides insight into the origins of the aerosol particles in the atmosphere. For instance, dust sources originating from northern Asia may be transported at higher elevations in the atmosphere, whereas local pollution is generally constrained closer to the surface. CALIOP further reveals the efficacy of the biomass burning injection height parameterization included in CM3 but not CM2.1, and shows that it is not sufficient. Comparing multiple aerosol optical properties derived by models to measurements from collocated instruments both identifies opportunities for the improvement of modeling aerosol distributions, as well as reveals important aspects governing aerosol properties. analyzing aerosol properties derived from

two related, but distinctly different global climate models is important in further determining how to improve the physics of the models for future versions. While further analysis is needed to pinpoint exactly how to improve the aerosol treatment in this model lineage, what is clear is that while some aspects have been drastically improved in the newer version CM3 (such as extinction magnitude and elevation) other aspects are worse than before (such as seasonality).

Key findings in this study include:

- All of the aerosol optical parameters in models should be evaluated against available observations in order to validate the model's credibility, as one parameter can reproduce satisfactorily the observations while another totally fails. This was demonstrated in Sect. 4.2.1, where for seven cities worldwide, the models accurately reproduced single-scattering albedo and the Ångström exponent, but under- or overestimated aerosol optical depth.

- Comparing AERONET data with CALIOP data shows the importance of measuring the vertical distribution of aerosols in the atmosphere. For example, while AERONET data shows that total AOD in February to May and September to November in Taiwan are similar, the CALIOP data reveals that the vertical distribution of the extinctions are considerably different during these two seasons. The difference in vertical distributions also provides insight into the origins of the aerosol particles in the atmosphere. For instance, dust sources originating from northern Asia may be transported at higher elevations in the atmosphere, whereas local pollution is generally constrained closer to the surface.

- Improvements to the modeling of aerosols originating from biomass burning sources involve increasing the injection height of biomass burning particles to avoid rapid removal by near-surface turbulence and therefore more properly represent the vertical profile of these species. Sect. 4.2.2 showed that model extinction vertical profiles over biomass burning regions, even in CM3 which accounted for the injection height, were too low as compared to AERONET and CALIOP data.

- Finally, analyzing aerosol properties derived from two related, but distinctly different global climate models is important in further determining how to improve the physics of the models for future versions. While further analysis is needed to pinpoint exactly how to improve the aerosol treatment in this model lineage, what is clear is that while some aspects have been drastically improved in the newer version CM3 (such as extinction magnitude and elevation) other aspects are worse than before (such as seasonality).

[revised manuscript text omitted]

---

## Author Response (AR2)

**RESPONSE TO THE COMMENTS OF THE REVIEWERS**

Manuscript Ref: acp-2016-790

**Comparing multiple model-derived aerosol optical properties to spatially collocated ground-based**

**and satellite measurements**

I. B. Ocko and P. A. Ginoux

We thank the Editor and referees for their time and for carefully reviewing our manuscript. Below we respond to comments by Reviewer #1.

1

**Responses to Reviewer #1:**

**Comment 1:** The authors have included some welcome additional figures to indicate model performance over wider regions of the globe. They have attempted to address my concerns on the conclusions from this research and restructured the summary. Regarding interpolation to station location, I understand that this may not be an issue in general as the variation between adjacent grid boxes may be relatively small, especially when averaging over long timescales. Therefore, for this work interpolation is likely a minor issue. However, in my experience it is good practice to interpolate, especially in the case of comparing at polluted locations, as not to introduce unnecessary bias.

**Response:** We are happy to hear that the reviewer is satisfied with the additional figures and the restructuring of the conclusions section. The reviewer's previous comments have undoubtedly made the paper stronger, and we are very appreciative of their time and thoughtfulness.

**Comment 2: Minor edits:**

Fig 3 - the legend says black circles but they appear gray.

Fig 5 - "within each panels" (remove 's')

pg12 ln 9 - "seas salt" should be "sea-salt"

pg12 ln15 & 17 - consider changing "damping" to "reducing".

pg12 ln18 - Change "We will now..." to the section number as the analysis does not happen immediately after the statement.

pg18 ln12 - extra space after "However,"

pg22 ln2-5 - Saying that the vertical profile is worse in CM3 than CM2.1 doesn't negate the fact that the seasonality is better in CM3. I think this sentence is unnecessary.

pg22 ln22 - "overestimate" rather than "over-"

pg23 - The final paragraph does not conclude well. Accounting for the temporal collocation of instruments should either be improved for this study or the statement removed. The final sentence is superfluous. Consider integrating other research ideas into relevant paragraphs within the conclusions. An adequate concluding remark, regarding the necessity to consider multiple parameters and make used of the new 3-D composition information from observations, will need adding to the previous paragraph.

**Response:** We have made the requested changes. The final paragraph now reads (lines 23.27-24.2): "
[revised manuscript text omitted]